# Organic anion transporter 1 is an HDAC4-regulated mediator of nociceptive hypersensitivity in mice

Christian Litke [1], Anna M. Hagenston [1], Ann-Kristin Kenkel [1], Eszter Paldy[2], Jianning Lu[2], Rohini Kuner [2] & Daniela Mauceri [1]✉

Persistent pain is sustained by maladaptive changes in gene transcription resulting in altered function of the relevant circuits; therapies are still unsatisfactory. The epigenetic mechanisms and affected genes linking nociceptive activity to transcriptional changes and pathological sensitivity are unclear. Here, we found that, among several histone deacetylases (HDACs), synaptic activity specifically affects HDAC4 in murine spinal cord dorsal horn neurons. Noxious stimuli that induce long-lasting inflammatory hypersensitivity cause nuclear export and inactivation of HDAC4. The development of inflammation-associated mechanical hypersensitivity, but neither acute nor basal sensitivity, is impaired by the expression of a constitutively nuclear localized HDAC4 mutant. Next generation RNA-sequencing revealed an HDAC4-regulated gene program comprising mediators of sensitization including the organic anion transporter OAT1, known for its renal transport function. Using pharmacological and molecular tools to modulate OAT1 activity or expression, we causally link OAT1 to persistent inflammatory hypersensitivity in mice. Thus, HDAC4 is a key epigenetic regulator that translates nociceptive activity into sensitization by regulating OAT1, which is a potential target for pain-relieving therapies.

[1] Department of Neurobiology, Interdisciplinary Center for Neurosciences (IZN), Heidelberg University, INF 366, 69120 Heidelberg, Germany. [2] Institute of Pharmacology, Heidelberg University, INF 366, 69120 Heidelberg, Germany. ✉email: mauceri@nbio.uni-heidelberg.de

Chronic pain is a debilitating condition that affects a conspicuous percentage of the worldwide population, resulting in tremendous yearly socio-economic costs[1]. Treatments are still, however, far from satisfactory for many patients. Thus, there is an unmet need for the uncovering of novel molecular and cellular players serving as fresh targets for future, more adequate, therapeutic interventions.

Steps have been made in understanding the signals mediating central sensitization, which is characterized by aberrant pain sensation to both noxious and non-noxious stimuli, but a clear understanding of this complex adaptive process is still lacking. The transition from acute to chronic pain is considered a pathological manifestation of neuronal plasticity in nociceptive pathways[2,3] and shares common molecular mechanisms with other adaptive processes of the central nervous system (CNS) such as memory[4,5]. Signal-regulated gene transcription has been identified as one of the key events enabling the maladaptive plasticity typically associated with chronic pain[6]. In recent years, mounting evidence linked epigenetic–mediated regulation of transcription to different forms of chronic pain[7–11], but a precise picture is still missing.

Within the heterogeneous group of epigenetic players, DNA methyltransferases and histone deacetylases (HDACs) are prominent mediators of adaptive processes in the CNS owing to their capacity to translate incoming synaptic activity into long-lasting transcriptional responses. HDACs catalyze the removal of acetyl groups from the lysine residues of histone and non-histone proteins and regulate gene transcription by changing chromatin structure and modifying the activity of interacting partners[12]. The several existing HDACs can be categorized into four major classes according to their structural and functional properties. There are three classes of canonical zinc-dependent HDACs where the second class can be further divided into two subclasses based on the functional properties of its members (class I: HDAC1, -2, -3, -8; class IIa: HDAC4, -5, -7, -9; class IIb: HDAC6, -10; class IV: HDAC11), and one class comprised of the nicotinamide adenine dinucleotide-dependent sirtuins (class III)[13]. The members of class IIa can shuttle between the cytoplasm and the nucleus in a signal-dependent manner, and their subcellular localization is pivotal in physiological and pathological adaptations[14–17]. In particular, in pyramidal neurons, synaptic activity promotes the nuclear export of class IIa HDACs and negatively impacts their nuclear deacetylase activity[15,16,18,19]. In other types of neuronal cells, this phenomenon is less characterized.

Recent evidence points towards the importance of histone acetylation and HDAC activity in different pathological pain states[20–23]. Due to a lack of precise pharmacological tools, however, it is not yet possible to draw definitive conclusions regarding the roles of distinct HDACs, and redundancy is a major caveat. Further, there is no information on the regulation in spinal cord circuits of a fundamental aspect modulating the activity of class IIa HDACs, namely their subcellular localization, and how its potential alteration may functionally contribute to chronic pain. Most importantly from a mechanistic viewpoint, the identity of genes underpinning central sensitization and controlled by class IIa HDACs in spinal cord neurons in pathological pain is still elusive.

In this study, we examined a number of HDACs and found the class IIa member HDAC4 to be specifically regulated by synaptic activity and persistent inflammatory pain in neurons of the spinal dorsal horn. We further discovered that activity-dependent changes in the subcellular localization of HDAC4 contribute to mechanical hypersensitivity in mouse models of nociceptive inflammation without affecting either basal or acute sensitivity. Transcriptome analysis revealed organic anion transporter 1 (OAT1), a multi-specific antiporter of anionic substrates and drugs predominantly known for its role in kidney functions[24], to be an HDAC4 target in the spinal dorsal horn. Further, functional experiments validated spinally-expressed OAT1 as a mediator of nociceptive hypersensitivity and confirmed its targetability for treating inflammatory pain. Thus, OAT1 provides a molecular link between nociceptive activity, epigenetic-regulated transcription, and spinal sensitization, and represents a potential therapeutic target for the pharmacological treatment of persistent inflammatory pain.

## Results

**Synaptic activity triggers the nuclear export of HDAC4 in spinal cord neurons.** Given the large number of HDACs, we first investigated whether activity-dependent subcellular shuttling of HDACs takes place in neurons of the spinal cord, and used this property as a screening tool to prioritize HDACs for further functional analyses. Following treatment of primary spinal cord neurons with the *gamma*-aminobutyric acid-A-receptor antagonist bicuculline (Bic) to induce bursts of synaptic activity, and consistent with our observations in hippocampal neurons[18], we found that members of class I (HDAC1 and 3), class IIb (HDAC10), and class IV (HDAC11) did not change their subcellular distribution (Supplementary Fig. 1a, Fig. 1b). Amongst members of class IIa, HDAC4 was unique in showing a significant reduction of its nuclear content as a result of increased synaptic activity, whereas the nuclear content of HDAC5, -7 and -9 was unaltered (Fig. 1a, b, Supplementary Fig. 1a). These observations stand in striking contrast to those made for pyramidal neurons, where all class IIa HDACs shuttle between the cytosol and the nucleus in an activity-dependent manner[15,16,18,19]. The expression levels of HDAC4 and of all tested HDACs remained constant (Supplementary Fig. 1b, c), suggesting that the detected activity-dependent reduction of nuclear HDAC4 (Fig. 1a, b) stems from nuclear export rather than degradation. Notably, protein levels of the immediate early gene c-Fos were increased in the same samples, indicating that the synaptic activity stimulation protocol was successful (Supplementary Fig. 1b, c). Moreover, Bic stimulation evoked repetitive, rhythmic nuclear calcium rises consistent with action potential bursting (Supplementary Fig. 1d, e)[25]. In line with a functional relevance for the redistribution of HDAC4 in cultured spinal cord neurons, synaptic activity induced a significant increase of acetylated histone 3-Lys9 (AcH3) levels whereas total histone 3 levels remained constant (Supplementary Fig. 1f, g). Immunocytochemistry revealed that the activity-dependent increase of AcH3 took place specifically in neurons and not in glial cells present in the cultures (Fig. 1c, d). Thus, synaptic activity promotes the nucleocytoplasmic export specifically of class IIa HDAC member HDAC4 and acetylation of histone 3 in spinal cord neurons ex vivo.

**Long-lasting inflammatory nociception induces the nuclear export of HDAC4 in spinal dorsal horn neurons in vivo.** We proceeded to investigate whether nociceptive activity affects the subcellular localization of HDAC4 in spinal cord neurons in vivo. We made use of a mouse model of long-lasting inflammatory sensitization induced by hindpaw intraplantar injection of Complete Freund's adjuvant (CFA), which is frequently used to mimic the time course of chronic inflammatory pain conditions such as rheumatoid arthritis or tendonitis[26]. CFA injection induced paw edema within 24 h that persisted at least seven days (Supplementary Fig. 2a, b) and led to mechanical and thermal hypersensitivity (Supplementary Fig. 2c, d). We analyzed the subcellular localization of class IIa HDACs and the levels of AcH3 specifically in the nucleus of dorsal horn neurons of the lumbar spinal cord segments L3-L5 at different timepoints (0.5 h,

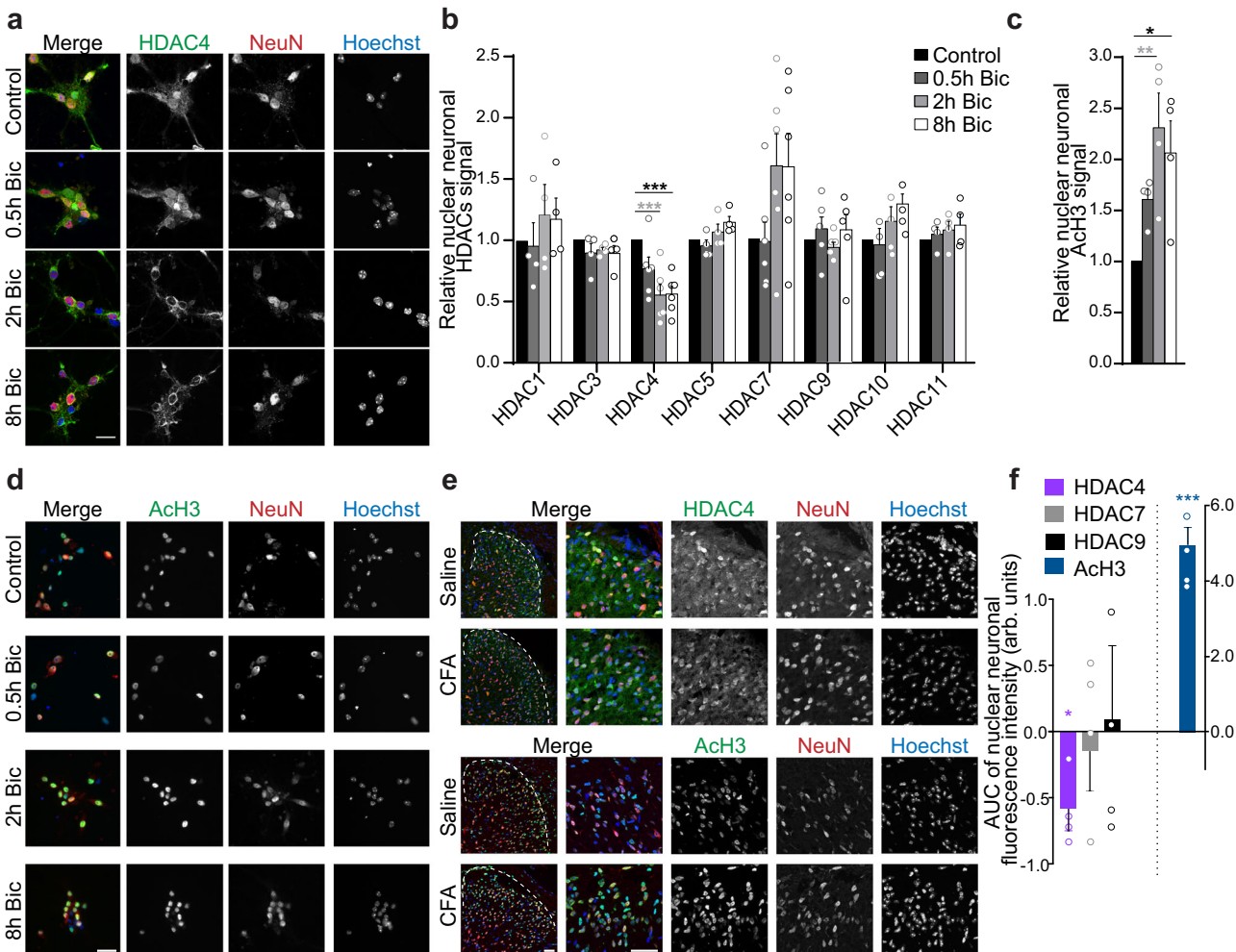

**Fig. 1 Neuronal activity and persistent inflammatory nociception trigger the nuclear export of HDAC4 and histone 3 acetylation in spinal cord neurons.**
**a** Representative images of cultured primary spinal cord neurons (DIV10) immunostained for endogenous HDAC4 (green) and the neuronal marker NeuN (red); Hoechst (blue) marks the nuclei. Scale bar is 20 μm. **b** Quantification of the relative fluorescence intensity of the nuclear HDACs signal in spinal cord neurons normalized to respective controls. Each point represents the mean value derived from one independent experiment. Ca. 40 cells were analyzed per condition and experiment (HDAC1 $n = 4$ $F_{(3,8)} = 1.120$; HDAC3 $n = 4$ $F_{(3,12)} = 0.9239$; HDAC4 $n = 6$ $F_{(3,20)} = 8.676$; HDAC5 $n = 4$ $F_{(3,12)} = 3.210$; HDAC7 $n = 6$ $F_{(3,20)} = 2.726$; HDAC9 $n = 5$ $F_{(3,20)} = 0.7916$; HDAC10 $n = 4$ $F_{(3,8)} = 2.502$; HDAC11 $n = 4$ $F_{(3,12)} = 0.6623$). **c** Quantification of the relative fluorescence intensity of the nuclear signal of acetylated histone H3-Lys9 (AcH3) in spinal cord neurons, treated as indicated. Ca. 30 cells were analyzed per sample ($n = 4$ $F_{(3,12)} = 5.854$). **d** Representative images of primary cultured spinal cord neurons (DIV10) immunostained for AcH3 (green). Cells were treated with Bic (50 μM) for the indicated times. Nuclei were labelled with Hoechst (blue) and NeuN (red) as a neuronal marker. Scale bar is 40 μm. **e** Representative images of spinal cord dorsal horn sections of lumbar spinal segments L3–5 of mice 24 h after intraplantar CFA or saline injection, immunostained for HDAC4 or acetylated histone H3-Lys9 (AcH3) shown in green; NeuN (red). Nuclei were labeled with Hoechst (blue) and scale bar is 40 μm. Higher magnifications of the upper laminae are shown in the right panels with a scale bar of 40 μm. **f** Quantifications of the average nuclear fluorescence intensities from immunolabeled HDACs and AcH3 in neurons of the spinal cord dorsal horn, normalized to values obtained from saline-injected mice and integrated over a time course series ($n = 4$ mice per condition and timepoint). Timepoints used were 0.5, 2, 6, and 24 h. Statistically significant differences were determined by one-way ANOVA followed by Dunnett's post hoc test (**b**, **c**) or two-tailed Student's $t$-test (**f**). ***$p < 0.001$; **$p < 0.01$; *$p < 0.05$. In bar graphs, each point represents a value derived from one independent culture or mouse. Graphs represent mean ± SEM. See also Supplementary Figs. 1 and 2.

2 h, 6 h, and 24 h) post-CFA or saline injection and calculated the variations over time from the saline-baseline. We observed a reduction of the nuclear HDAC4 content in dorsal horn neurons of inflamed mice relative to saline-treated mice (Fig. 1e, f; Supplementary Fig. 2e–g). Consistent with our observations in cultured spinal cord neurons, the nociceptive input-triggered subcellular redistribution of HDACs appeared to be specific for HDAC4 as the nuclear content of other class IIa HDACs (HDAC7 and -9) was not affected (Fig. 1f, Supplementary Fig. 2g, h). Differential analysis between superficial and deeper laminae revealed that the reduction of nuclear HDAC4 in dorsal horn

neurons took place preferentially in laminae I-II (Supplementary Fig. 2g). Nuclear levels of HDAC4 in the ipsilateral dorsal spinal cord of mice subjected to the spared nerve injury (SNI) model of neuropathic pain did not change relative to the contralateral side (Supplementary Fig. 2i), suggesting that HDAC4 nuclear exclusion might be modulated differentially in distinct pain states. As expected, c-Fos levels and the number of c-Fos positive cells in the dorsal horn were significantly elevated 2 h after intraplantar CFA injections (Supplementary Fig. 2j–m)[6,27]. NF-kB is a transcription factor that regulates numerous inflammatory responses[28]. Expression of IkBα, an inhibitor of NF-kB that binds

to NF-kB and traps it in the cytoplasm[29], was significantly reduced 24 h after CFA treatment (Supplementary Fig. 2l, m). Total spinal dorsal horn levels of HDAC4 and other analyzed HDACs, however, were unchanged following paw inflammation (Supplementary Fig. 2l, m). Importantly, nuclear neuronal AcH3 levels were increased after intraplantar CFA injections (Fig. 1e, f, Supplementary Fig. 2g). In contrast to the CFA model of long-lasting inflammatory sensitization, intraplantar injection of capsaicin results in short-lasting, acute, and local inflammation[26]. Capsaicin did not trigger the induction of c-Fos, nor did it affect the levels of AcH3 (Supplementary Fig. 2n, o). Taken together, our analyses suggest that persistent, but not acute, inflammatory nociceptive activity induces the nuclear export of HDAC4 and elevates histone H3 acetylation in dorsal horn neurons of the spinal cord.

**Nuclear HDAC4 in spinal cord neurons alters the expression of the pain-relevant gene *Ptgs2* in vivo.** The subcellular localization of HDAC4 is modulated by activity (Fig. 1)[15,18,19]. Via changes in its localization, HDAC4 influences activity-dependent gene transcription in neurons, thereby playing an essential role in information processing and in different forms of plasticity[15–18,30]. We hypothesized that nociceptive activity-mediated nuclear exclusion of HDAC4 might be instrumental to central sensitization via the modulation of gene transcription in spinal cord neurons. To test this hypothesis, we employed a constitutively nuclear localized dominant-active mutant of HDAC4 (HDAC4 3SA), which features three mutations in phosphorylation sites key for the nuclear export of HDAC4 and is therefore always confined to the nuclear compartment[16,18,19,31]. Overexpression of β-galactosidase (LacZ) as a control, wild-type HDAC4 (HDAC4 wt), or HDAC 3SA transgenes in cultured spinal cord neurons had no effect on neuronal viability (Supplementary Fig. 3a). We generated recombinant adeno-associated viruses (rAAVs) to express LacZ, HDAC4 wt, or HDAC4 3SA in dorsal horn neurons of the lumbar spinal cord in vivo (Fig. 2a). rAAV-injected spinal cord tissue revealed robust transgene expression at 3 weeks post-injection within the dorsal horn, whereas corresponding primary afferent neurons displayed no signs of viral infection (Supplementary Fig. 3b). Immunolabeling confirmed that the HDAC4 3SA mutant, in contrast to the HDAC4 wt version, remained localized in the nucleus 24 h after paw injection of CFA (Fig. 2b). In striking contrast to mice expressing LacZ or HDAC4 wt, CFA-triggered ipsilateral induction of AcH3 in dorsal horn neurons was prevented in mice expressing HDAC4 3SA (Fig. 2c, d), showing the importance of nociceptive activity-induced nuclear exclusion of HDAC4 for histone acetylation. Persistent nociceptive input from the periphery elicited by hindpaw CFA injection modulates the transcription of several plasticity-related genes in dorsal horn neurons, including genes encoding important effectors of nociceptive hypersensitivity such as *Ptgs2*, a critical mediator of inflammatory hypersensitivity[6,32,33] that has been shown to depend on HDAC4[18]. mRNA levels of *Ptgs2* were significantly induced in the dorsal lumbar segments of mice expressing either LacZ or HDAC4 wt 24 h after CFA injection[6], but not in mice expressing HDAC4 3SA (Fig. 2e). In summary, these data suggest that the shuttling of HDAC4 towards the cytosol results in histone H3 acetylation and the modified transcription of pain-related genes in spinal neurons.

**Nuclear HDAC4 in dorsal horn neurons reduces long-lasting nociceptive inflammatory hypersensitivity but does not affect acute or basal nociception.** We next sought to investigate the relevance of the activity-induced nuclear exclusion of HDAC4 for

acute nociception and for inflammation-associated nociceptive hypersensitivity. Mice intra-spinally injected with rAAV-HDAC4 3SA and controls showed similar basal sensitivity to mechanical and thermal stimuli applied to the hindpaws (Fig. 2f–h). Following CFA delivery, all mice developed mechanical and thermal hypersensitivity lasting up to 10 days (Fig. 2f–h)[6,34]. Thermal sensitivity was similar in all experimental groups (Fig. 2f). However, in comparison to control mice, mice expressing HDAC4 3SA in dorsal horn neurons showed significantly reduced mechanical hypersensitivity starting 24 h after CFA injection and persisting until 6 days (Fig. 2g, h). Further, HDAC4 3SA-expressing mice showed drastically reduced allodynia, which is characterized by withdrawal responses to typically non-noxious intensities of force such as 0.04 g or 0.07 g (Fig. 2i, j), indicating that constitutively nuclear HDAC4 opposes the spinal sensitization associated with persistent inflammatory mechanical hypersensitivity. In contrast to these findings, acute nocifensive behavior elicited within five minutes after hindpaw injection of capsaicin was not different between mice expressing HDAC4 3SA in the dorsal spinal cord and those expressing LacZ or HDAC4 wt (Fig. 2k). Analysis of the typical bi-phasic nociceptive response following intraplantar injection of formalin in mice expressing HDAC4 3SA or controls also revealed no significant differences between experimental groups in the durations of nocifensive behaviors during the acute phase (0–10 min after injection) (Fig. 2l, m). During the second phase (10–60 min after injection), however, which is characterized by both on-going nociceptor activation and plasticity in central neurons[35–37], mice expressing HDAC4 3SA displayed significantly reduced nocifensive behaviors compared to controls (Fig. 2l, m). Thus, nuclear HDAC4 in dorsal horn neurons inhibits the development of persistent inflammatory spinal sensitization without altering basal sensitivity or acute nociception.

**HDAC4 shapes the transcriptional profile of the spinal cord in persistent inflammatory sensitization.** We next addressed the mechanistic basis of how nuclear HDAC4 shapes inflammatory pain by performing a transcriptome analysis using next generation RNA sequencing (RNAseq) of dorsal spinal tissue from mice spinally expressing LacZ, HDAC4 wt, or HDAC4 3SA. 24 h following CFA injection, 104 genes were upregulated and nine genes downregulated in the spinal dorsal horn of LacZ-expressing mice, 47 genes were upregulated and five genes downregulated in HDAC4 wt-expressing mice, and 252 genes were upregulated and six genes downregulated in mice expressing HDAC4 3SA (Fig. 3a, c; Supplementary Table 1). The biological processes identified using gene ontology (GO) term analysis of differentially expressed genes in LacZ-expressing mice pointed towards an inflammatory immune response in CFA-injected mice, as expected (Fig. 3b). Comparisons of CFA-mediated changes in the expression levels of the differentially expressed genes (DEGs) between mice spinally expressing LacZ, HDAC4wt, and HDAC4 3SA revealed striking differences (Fig. 3d). Since we wanted to dissect the specific role of HDAC4 subcellular localization in sensitization (Fig. 2), we focused our attention on those DEGs sexhibiting differentially regulated CFA-triggered expression changes in HDAC4 3SA-expressing mice as compared to both LacZ- and HDAC4 wt-expressing mice (a selection is shown in Fig. 3e). This pool of DEGs included targets with reported functions in pain, such as thrombospondin 1 (*Thbs1*), a secreted glycoprotein involved in inflammatory responses;[38–40] *H19*, a long non-coding RNA recently linked to the development of neuropathic pain;[41] myelin protein zero (*Mpz*), which has been linked to neuropathy;[42] and periaxin (*Prx*), which is associated with

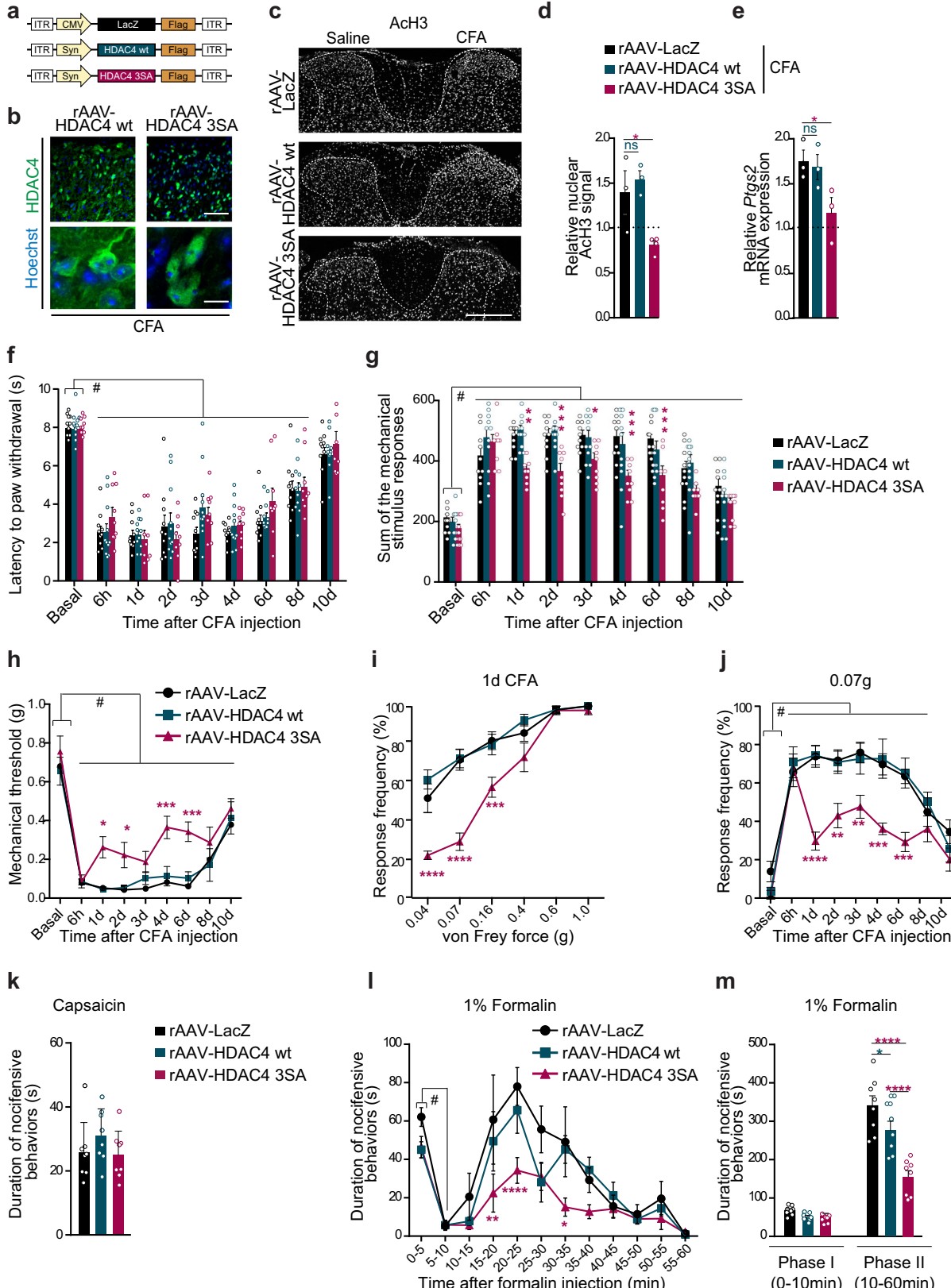

neuropathic pain[43]. Changes in the expression levels of selected genes were confirmed in an extended cohort of samples using quantitative RT-PCR (QRT-PCR) analysis (Fig. 3f). These data thus identify an inflammatory pain-related transcriptional program in the dorsal spinal cord controlled by the subcellular localization of HDAC4.

**Expression of the HDAC4 target OAT1 in the spinal cord modulates inflammatory hypersensitivity.** Among the genes whose induction by CFA in control conditions was strongly impaired by constitutively nuclear HDAC4, our attention was drawn to *Slc22a6*, which codes for the organic anion transporter 1 (OAT1), also known as solute carrier family 22 member 6. OAT1,

**Fig. 2 Nuclear HDAC4 alters the expression of pain-relevant genes in vivo and reduces hypersensitivity in persistent but not acute inflammatory pain models. a** Schema of the rAAVs used. **b** Images of dorsal horn from spinal cord sections L3–5 following intraspinal injections of rAAV-HDAC4 wt or -HDAC4 3SA and intraplantar CFA injection. Constructs (green), Hoechst (blue, nuclei). Scale bar is 100 µm in the upper panels and 10 µm in the lower panels. **c** Images of dorsal sections of spinal cord L3–5, immunostained for acetylated histone H3-Lys9 (AcH3), intra-spinally injected as indicated, followed by injections of CFA or saline for 24 h. Scale bar is 300 µm. **d** Relative fluorescence intensities of the nuclear AcH3 signal in neurons of the dorsal horn following intraspinal rAAV delivery and intraplantar injections as in (**c**), normalized to contralateral (saline) ($n = 3$). **e** QRT-PCR analysis of *Ptgs2* in lumbar L3–5 spinal cord segments 24 h after injections, in mice expressing indicated constructs ($n = 3$). Values were normalized to rAAV-LacZ saline-injected mice. **f** Paw withdrawal latency to a radiant heat light source. **g** Mechanical sensitivity following CFA injection. **h** 40% response threshold to mechanical force via von Frey filaments. **i** Stimulus intensity-response frequency curve for plantar von Frey mechanical stimulation for the same cohort of mice shown in (**f–h**) 24 h after CFA injection. **j** Response frequency to light touch (0.07 g filament) over time (LacZ $n = 10$, HDAC4 wt $n = 11$, HDAC4 3SA $n = 9$, **f–j**). **k** Cumulative durations of acute nocifensive behaviors evoked by capsaicin ($n = 8$). **l**, **m** Cumulative durations of nocifensive behaviors evoked by formalin. Bar graphs in **m** represent the sum of all nocifensive behaviors in the early (Phase I) and late (Phase II) phases, and panel **l** shows the response curves over time (LacZ, HDAC4 3SA $n = 8$; HDAC4 wt $n = 7$). Statistically significant differences were determined by one-way ANOVA followed by Dunnett's post hoc test (**d**, $F_{(2,6)} = 0.8689697$) or Tukey's test (**k**, $F_{(2,21)} = 1.220$); two-way ANOVA followed by Dunnett's post hoc test (**e**; $F_{(1,11)} = 44.47$) or two-way ANOVA with repeated measures followed by Dunnett's post hoc test for comparisons to basal values and Tukey's post hoc test for comparisons between conditions (**f**, $F_{(8,216)} = 83.08$ F $_{(2,27)} = 0.9383$; **g**, $F_{(8,216)} = 66.30$ $F_{(2,27)} = 10.47$; **h**, $F_{(8, 216)} = 49.47$ $F_{(2,27)} = 17.48$; **i**, $F_{(5,135)} = 88.67$ $F_{(2,27)} = 24.19$; **j**, $F_{(8,216)} = 37.44$ $F_{(2,27)} = 16.61$; **l**, $F_{(11,220)} = 15.24$ $F_{(2,20)} = 17.70$; **m**, $F_{(1,20)} = 243.6$ $F_{(2,20)} = 17.25$). ****$p < 0.0001$; ***$p < 0.001$; **$p < 0.01$; *$p < 0.05$; #$p < 0.0001$. Asterisks (*) refer to statistical comparisons between conditions and hashtags (#) to comparisons relative to basal values within each condition. Each point represents the mean value derived from one mouse. Graphs represent mean ± SEM. See also Supplementary Fig. 3.

which is primarily expressed in the kidney, has also been found in the CNS and is known to mediate the transport of diverse low molecular weight endogenous and exogenous substrates, including steroids, hormones, neurotransmitters, numerous drugs, and xenobiotics[24,44]. However, a detailed understanding of its function in the nervous system is still lacking. In agreement with our data showing increased mRNA levels of *Slc22a6* after intraplantar CFA injection (Fig. 3e, f), immunoblot analysis of dorsal spinal cord (L3–5) tissue of mice injected with CFA 0.5, 2, 6, or 24 h prior to harvest showed a significant increase in OAT1 protein levels at 24 h (Fig. 4a, b). Immunohistochemical analyses confirmed that OAT1 expression increased in the dorsal horn (Fig. 4c, d). Further, co-localization analysis showed that the numbers of OAT1-positive neurons, but not the numbers of OAT1-positive non-neuronal cells were increased 24 h after CFA injection, indicating that the induction of OAT1 in persistent inflammatory nociception takes place primarily in neurons (Fig. 4e–g). This conclusion was further validated by immunohistochemical analysis showing no readily detectable OAT1 in astrocytes and microglia in the dorsal horn (Supplementary Fig. 4). As was the case for HDAC4 nucleus-to-cytoplasm translocation, expression levels of OAT1 in the ipsilateral dorsal spinal cord of mice subjected to the SNI model of neuropathic pain did not change relative to the contralateral side (Fig. 4h), suggesting that, like HDAC4 localization, OAT1 expression might be modulated differentially in distinct pain states. In keeping with our RNAseq analysis, immunohistochemical analyses of the spinal dorsal horn from mice intra-spinally injected with rAAV-LacZ, -HDAC4 wt, or -HDAC4 3SA showed that nuclear HDAC4 prevented the CFA-triggered induction of OAT1 (Fig. 4i). As OAT1 expression in the kidney differs between male and female rodents[45], and the bulk of our analyses thus far were carried out in male mice, we sought out to determine whether an HDAC4-dependent regulation of OAT1 would also take place in female mice. We found that 24 h after CFA delivery, nuclear levels of HDAC4 in dorsal horn neurons of adult female mice were reduced in a manner similar to what we previously detected for male mice (Fig. 4j). Further, relative to saline-injected controls, female mice that received a CFA injection exhibited an induction of OAT1 expression in the dorsal horn (Fig. 4k). Finally, in agreement with the observations we made using male mice (Fig. 2), dorsal spinal expression of HDAC4 3SA in female mice prevented the CFA-mediated induction of both mechanical hypersensitivity (Fig. 4l) and OAT1 expression (Fig. 4m).

Thus, persistent inflammatory nociceptive activity, by promoting the exclusion of HDAC4 from the nucleus, derepresses the expression of OAT1 in spinal dorsal horn neurons in both male and female mice.

To test the functional implications of this finding, we initially used an acute approach involving the intrathecal delivery of siRNAs targeting *Slc22a6* (siOat1) to achieve knockdown of OAT1 in the dorsal spinal cord (Fig. 5a). The efficacy of siRNA treatment for reducing *Slc22a6* expression in the spinal dorsal horn in vivo was confirmed (Supplementary Fig. 5a). Mice that received intrathecal siOat1 treatment showed significantly reduced CFA-triggered mechanical hypersensitivity and allodynia 6 h and 24 h after CFA injection, but exhibited no change in basal sensitivity or thermal hyperalgesia (Fig. 5b–d; Supplementary Fig. 5b, c). Mechanical sensitivity in the uninflamed saline-treated contralateral hindpaw also did not change following delivery of siOat1 (Supplementary Fig. 5d).

Next, to achieve longer-lasting and spatially restricted reduction of OAT1 expression in the spinal cord dorsal horn, we generated rAAV vectors containing DNA sequences encoding short hairpin RNAs (shRNAs) designed to target two different regions of the *Slc22a6* mRNA sequence (rAAV-shOat1-1 and -shOat1-2). We verified their efficiency for reducing *Slc22a6*/OAT1 mRNA and protein levels in cultured neurons (Supplementary Fig. 5e, f). Bilateral intraspinal delivery of either rAAV-shOat1-1 or rAAV-shOat1-2 three weeks prior to intraplantar CFA injection (Fig. 5e) significantly attenuated CFA-induced mechanical hypersensitivity and allodynia compared to mice intra-spinally injected with a control rAAV encoding an shRNA not targeting any gene (rAAV-shUNC; Fig. 5f–h; Supplementary Fig. 5g). Neither basal sensitivity nor thermal hypersensitivity (Fig. 5f, g; Supplementary Fig. 5g, h), nor sensitivity in the uninflamed contralateral side (Supplementary Fig. 5i), nor capsaicin-induced acute inflammatory nocifensive behaviors (Fig. 5i) were affected by rAAV-shOat1 delivery. At the end of the behavioral experiments, we confirmed the in vivo knockdown of OAT1 expression (*Scl22a6*, Supplementary Fig. 5j–l).

To understand whether, conversely, elevated levels of OAT1 in the spinal dorsal horn could be sufficient to increase sensitivity and exacerbate inflammatory nociceptive sensitivity, we overexpressed OAT1 in the dorsal horn of adult mice via rAAVs (rAAV-OAT1). As a control, we injected rAAV-LacZ. We verified in cultured neurons and in the dorsal horns of injected mice the correct expression of transgenes (Supplementary Fig. 6a–d). Compared to LacZ-overexpressing controls, and

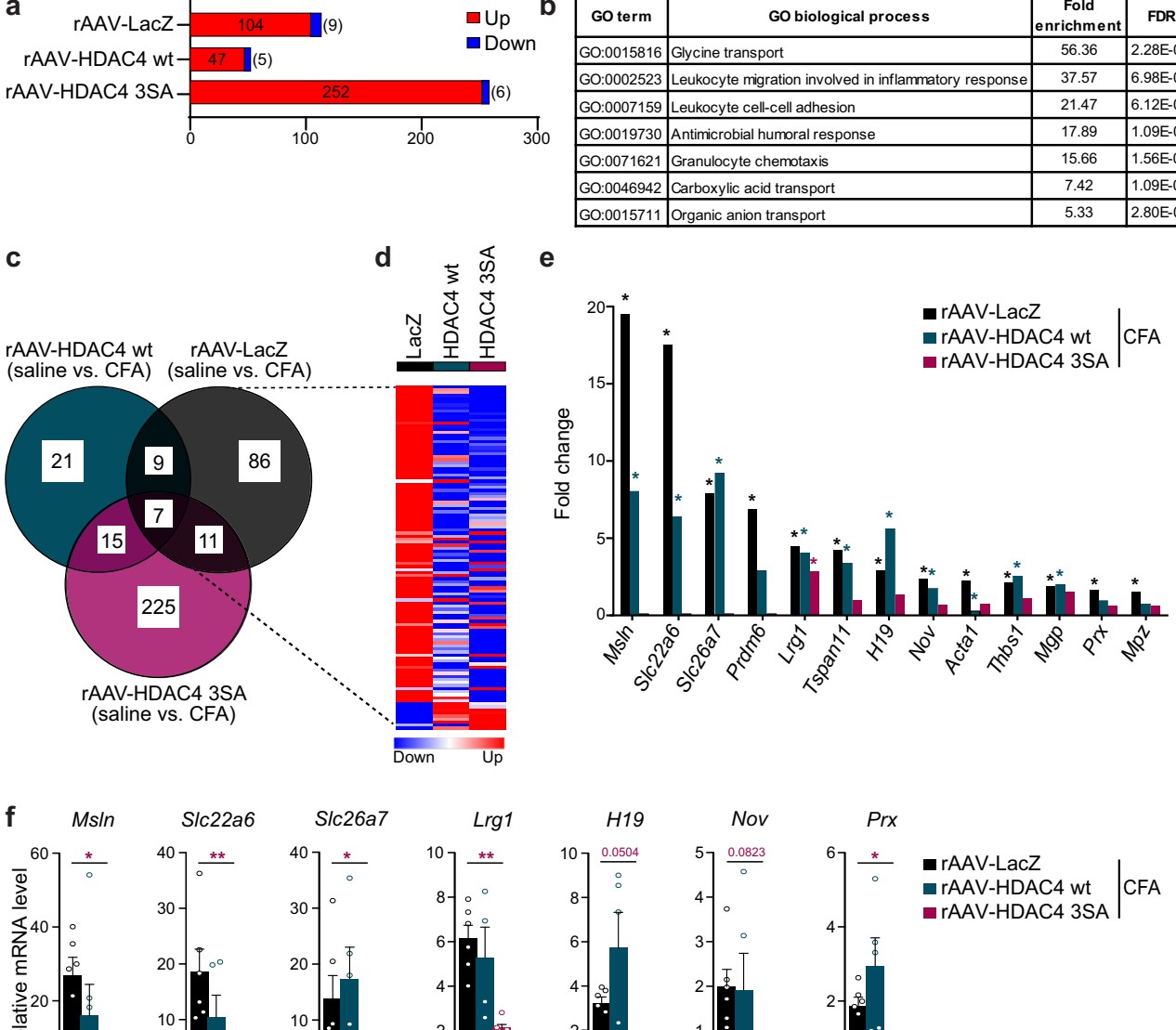

**Fig. 3 HDAC4 regulates gene expression in persistent inflammatory sensitization in the spinal cord. a** Number of differentially expressed genes (DEGs) ($p_{adjusted} < 0.05$), identified by RNAseq analysis, in the dorsal spinal cord (L3–5) of mice intra-spinally injected with either rAAV-LacZ, -HDAC4 wt, or -HDAC4 3SA that are significantly up- or downregulated following intraplantar injection of CFA ($n = 3$ mice/group). **b** Gene ontology (GO) analysis of DEGs that are significantly up- or downregulated following CFA injection in LacZ-expressing mice, using the PANTHER overrepresentation test and database. Enriched GO terms with an enrichment score >5 are listed. **c** Venn diagram illustrating the number and overlap of significantly up- or downregulated DEGs following CFA injection, as in (**a**). **d** Heat map of the relative fold change of significantly DEGs in LacZ-expressing mice after CFA injection compared to the fold changes of the same genes in rAAV-HDAC4 wt or -HDAC4 3SA injected mice. Color-coding is adjusted per row. **e** Relative expression fold changes of a selection of CFA-induced genes identified in the RNAseq analysis that are not induced in HDAC4 3SA-expressing mice. Asterisks refer to the statistical values provided in Supplementary Table 1. **f** Relative expression fold changes from QRT-PCR validation analysis using additional samples (*Msln, Slc22a6, Slc26a7, Lrg1, H19, Nov*, and *Prx*; $n = 6$ mice per condition). Statistically significant differences were determined by one-way ANOVA followed by Dunnett's post hoc test (*Msln*, $F_{(2, 14)} = 4.563$; *Slc22a6*, $F_{(2, 14)} = 8.223$; *Slc26a7* $F_{(2, 14)} = 5.243$; *Lrg1*, $F_{(2, 13)} = 9.750$; *H19*, $F_{(2, 13)} = 7.209$; *Nov*, $F_{(2, 14)} = 2.960$; *Prx*, $F_{(2, 14)} = 8.688$). Each point represents the mean value derived from one mouse. Bars represent mean + SEM. See also Supplementary Table 1.

similar to what we had observed in uninfected mice that received an intraplantar CFA injection (Supplementary Fig. 2l, m), overexpression of OAT1 in the dorsal spinal cord significantly decreased IkBα levels (Supplementary Fig. 6d, e). Three weeks after bilateral intraspinal rAAV delivery (Fig. 5e), basal mechanical sensitivity was significantly increased in OAT1-overexpressing mice compared to control mice expressing LacZ

(Fig. 5j, k; Supplementary Fig. 6f). Further, in the CFA-model, mice intra-spinally injected with rAAV-OAT1 developed consistently more intense mechanical hyperalgesia and allodynia than controls in both CFA-treated (ipsilateral) and saline-treated (contralateral) paws (Fig. 5j–l; Supplementary Fig. 6g, h). In contrast, capsaicin-induced acute nocifensive behavior was not affected by OAT1 overexpression (Fig. 5m). Taken together, these

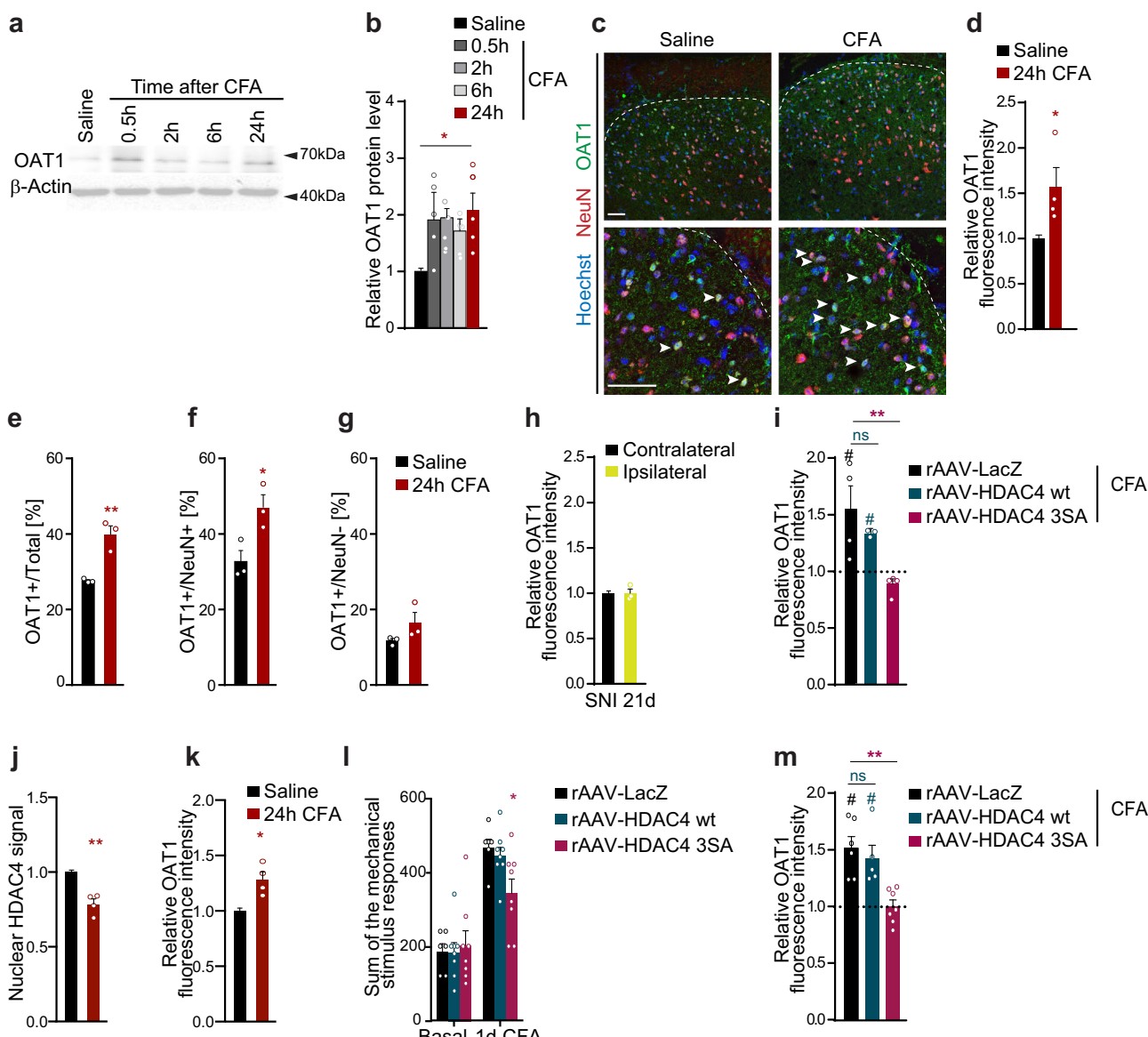

**Fig. 4 Activity of the HDAC4 target OAT1 in the spinal cord mediates mechanical hypersensitivity. a** Immunoblots of OAT1 and β-Actin from dorsal spinal cord segments L3–5. **b** Relative protein levels of OAT1 and β-Actin normalized to values from saline-injected control mice ($n = 5$). **c** Representative images of dorsal horn sections of spinal segments L3–5 of mice 24 h after injection, OAT1 (green), NeuN (red, neuronal marker), Hoechst (blue, nuclear marker). Lower panels are a higher magnification. Arrows indicate OAT1-positive and NeuN positive cells. Scale bars are 40 μm. **d** Relative OAT1 immunofluorescence intensity in dorsal horn following injections as in (**c**) ($n = 4$). **e–g** Co-localization analysis of OAT1-positive cells in the dorsal horn following injections 24 h earlier. Graph represents the percentage of OAT1-positive cells within the total cell population (**e**, Hoechst-positive), within the neuronal population (**f**, NeuN-positive), and within non-neuronal cells (**g**, NeuN-negative) ($n = 3$). **h** Relative OAT1 immunofluorescence intensity in dorsal horn 21 days after SNI, normalized to the contralateral side ($n = 3$). **i** Relative OAT1 immunofluorescence intensity in dorsal horn laminae I–V of mice intra-spinally injected as indicated, followed by intraplantar injections of CFA or saline for 24 h (LacZ, HDAC4 3SA $n = 4$; HDAC4 wt $n = 3$). **j, k** Relative fluorescence intensities of the nuclear HDAC4 (**j**) and OAT1 (**k**) signal in neurons of the dorsal horn from female mice, following intraplantar injections of CFA or saline for 24 h, normalised to contralateral (saline) sides ($n = 4$). **l** Mechanical sensitivity of female mice intra-spinally injected as indicated, followed by unilateral intraplantar injections of CFA or saline for 24 h (**l, m**; LacZ = 7; HDAC4 wt $n = 6$; HDAC4 3SA $n = 8$). **m** Relative OAT1 immunofluorescence intensity in dorsal horn laminae I–V of female mice intra-spinally injected as in (**l**), followed by intraplantar injections of CFA or saline for 24 h prior to tissue harvest ($n = 5–7$). Statistically significant differences were determined by two-tailed Student's $t$-test (**d–h, j, k**), one-way ANOVA followed by Dunnett's post hoc test (**b**, $F_{(4,20)} = 4.281$; **m**, $F_{(2, 15)} = 10.567$), or two-way ANOVA followed by Dunnett's post hoc test (**i**, $F_{(2,8)} = 9.016$; **l**, $F_{(2, 19)} = 1.087$). ** $p < 0.01$; #, * $p < 0.05$. Asterisks (*) refer to statistical comparisons between conditions and hashtags (#) to comparisons relative to basal values within each condition. Each point represents the mean value derived from one mouse. Graphs represent mean + SEM. See also Supplementary Fig. 4.

experiments indicate that alterations in OAT1 expression in spinal cord dorsal horn neurons can significantly influence the development of nociceptive hypersensitivity, and as such provide a mechanistic molecular link between epigenetic modulation of gene expression and persistent inflammatory sensitization.

**Pharmacological inhibition of OAT1 reduces established long-lasting inflammatory hypersensitivity.** To explore the translational potential of our findings, we searched for FDA-approved inhibitors of OAT1 activity and found that the uricosuric drug Probenecid (PBN) is, at present, the sole OAT1 inhibitor suitable for

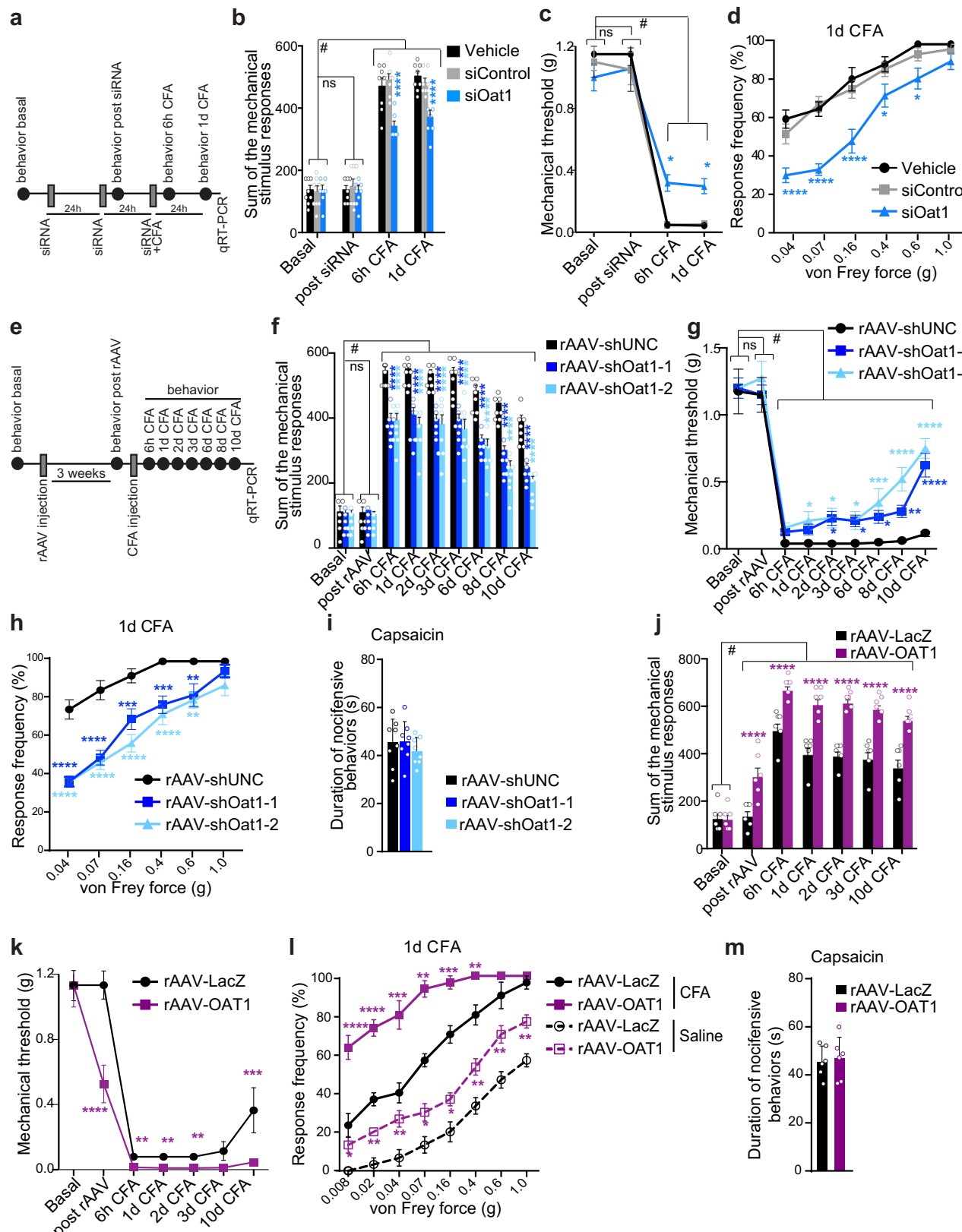

clinical use (https://transportal.compbio.ucsf.edu/transporters/SLC22A6/#inhib). PBN is largely used in clinical practice for the treatment of gout and hyperuricemia[24,46]. In terms of specificity, the IC50 of PBN for OAT1 is reported to be at least one order of magnitude smaller than for other potential targets[47,48]. We delivered PBN or vehicle intrathecally to preferentially target OAT1 in the

nervous system of inflamed mice. Compared to vehicle-treated inflamed mice, mice treated with PBN displayed significantly reduced CFA-triggered mechanical allodynia up to 24 h after a single intrathecal dose (Fig. 6a–c). PBN had no influence on basal mechanical responses or on the development of CFA-induced thermal hyperalgesia (Fig. 6d). Thus, PBN inhibits the development

**Fig. 5 OAT1 is a functional mediator of inflammatory hypersensitivity. a** Timeframe of the experiment. **b** Sum of the stimulus-evoked responses to von Frey filaments. **c** 40% response threshold to mechanical force via von Frey filaments. **d** Stimulus intensity-response frequency curve for plantar von Frey mechanical stimulation 24 h after intraplantar CFA and final intrathecal siRNA injection (**b–d**, vehicle, siControl $n = 8$; siOat1 $n = 7$). **e** Timeframe of the experiment and **f** mechanical sensitivity following intraplantar CFA injection, represented as the average sum of the von Frey stimulus responses. **g** 40% response threshold to mechanical force via von Frey filaments. **h** Stimulus intensity-response frequency curve for plantar von Frey mechanical stimulation 24 h post intraplantar CFA (**f–h**, $n = 8$). **i** Cumulative durations of acute nocifensive behaviors evoked by intraplantar injection of capsaicin for mice intra-spinally injected as indicated ($n = 8$). **j** Mechanical hypersensitivity represented as the average sum of the von Frey stimulus responses. **k** 40% response threshold to mechanical force via von Frey filaments. **l** Stimulus intensity-response frequency curve for plantar von Frey mechanical stimulation 24 h after intraplantar CFA injection (**j–l**, $n = 6$). **m** Cumulative durations of acute nocifensive behaviors evoked by intraplantar injection of capsaicin for mice intra-spinally injected as indicated ($n = 6$). Statistically significant differences were determined by two-way ANOVA with repeated measures followed by Dunnett's post hoc test (**b**, $F_{(3,60)} = 461.2$ $F_{(2,20)} = 8.555$; **c**, $F_{(3,60)} = 177.7$ $F_{(2,20)} = 1.372$; **d**, $F_{(3,100)} = 98.07$ $F_{(2,20)} = 16.98$; **f**, $F_{(8,168)} = 429.9$ F $_{(2,21)} = 33.16$; **g**, $F_{(8,168)} = 152.3$ $F_{(2,21)} = 9.775$; **h**, $F_{(5,105)} = 103.7$ $F_{(2,21)} = 24.95$); one-way ANOVA followed by Tukey's post hoc test (**i**, $F_{(2,21)} = 0.5614$); two-way ANOVA with repeated measures followed by Dunnett's post hoc test and multiple $t$-test with Holm-Sidak correction (**j**, $F_{(6,60)} = 241.1$ $F_{(1,10)} = 34.17$; **k**, $F_{(6,60)} = 88.36$ $F_{(1,10)} = 6.260$; **l**, $F_{(7,140)} = 168.6$ $F_{(3,20)} = 92.58$); or **m**, two-tailed Student's $t$-test. ****$p < 0.0001$; ***$p < 0.001$; **$p < 0.01$; *$p < 0.05$; #$p < 0.0001$. Asterisks (*) refer to statistical comparisons between conditions and hashtags (#) to comparisons relative to basal values in each condition. In bar graphs, each point represents the mean value derived from one independent mouse. Graphs represent mean ± SEM. See also Supplementary Figs. 5 and 6.

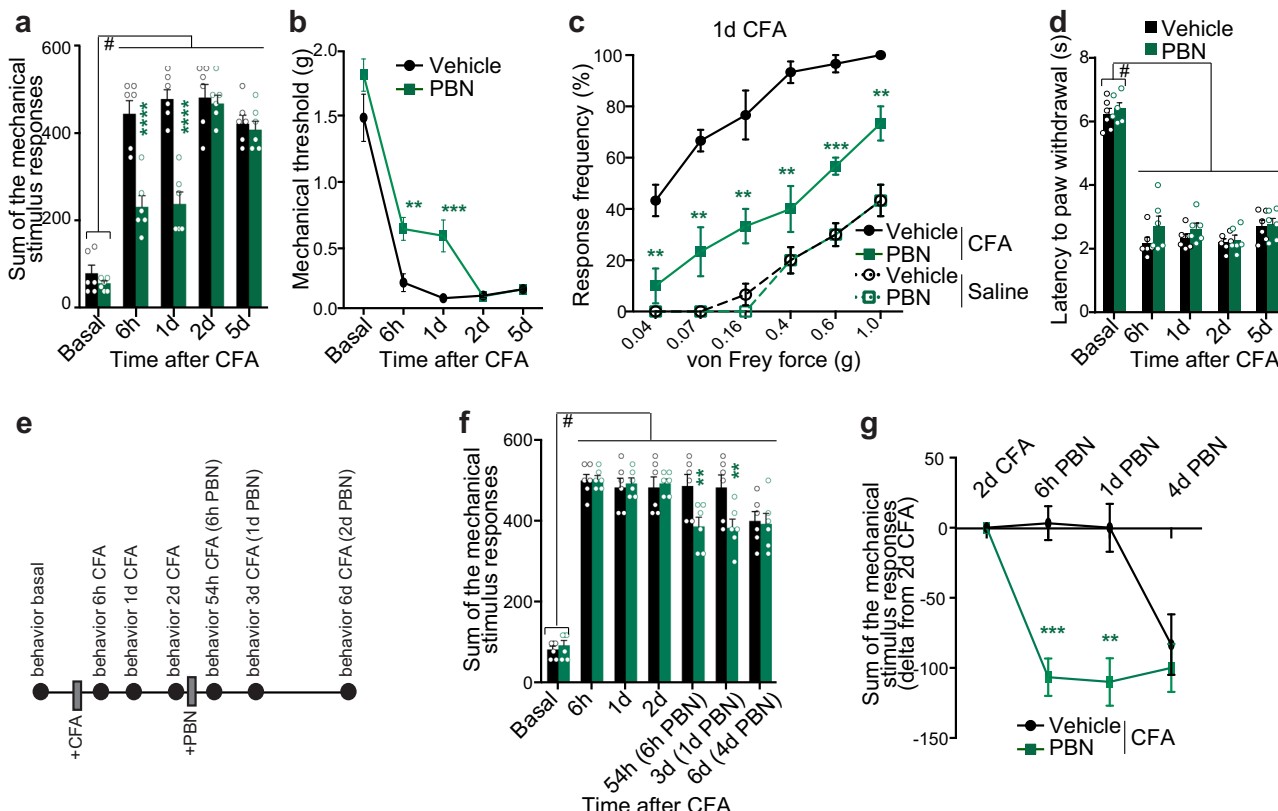

**Fig. 6 Inhibition of OAT1 reduces inflammatory hypersensitivity. a** Mechanical sensitivity following intraplantar CFA injection, represented as the sum of stimulus-evoked responses to von Frey filaments. **b** 40% response threshold to mechanical force via von Frey filaments. **c** Stimulus intensity-response frequency curve for plantar von Frey mechanical stimulation for the same cohort of mice shown in (**a**, **b**) 24 h after intraplantar CFA injection. **d** Paw withdrawal latency to a radiant heat light source for all filaments (**a–d**, $n = 6$). **e** Timeframe of the experiment. **f** Sum of the stimulus-evoked responses to von Frey filaments at the indicated timepoints for mice that received intrathecal injection of probenecid (PBN) or vehicle two days after CFA delivery. **g** Sum of mechanical stimulus-evoked responses at the indicated timepoints for the same cohort of mice as in (**f**), calculated as the difference relative to values obtained on the day of intrathecal injection of PBN or vehicle. (**f**, **g**, $n = 6$). Statistically significant differences were determined by two-way ANOVA with repeated measures followed by Dunnett's post hoc test for comparisons to basal values and multiple $t$-test with Holm-Sidak correction (**a**, F $_{(4,40)} = 229.2$ F $_{(1,10)} = 14.48$; **b**, $F_{(4,40)} = 137.6$ $F_{(1,10)} = 9.686$; **c**, $F_{(5,100)} = 73.10$ $F_{(3,20)} = 82.41$; **d**, $F_{(4,40)} = 182.1$ $F_{(1,10)} = 2.408$; **f**, $F_{(6,60)} = 247.5$ $F_{(1,10)} = 1.062$; **g**, $F_{(3,30)} = 20.21$ $F_{(1,10)} = 17.16$). ****$p < 0.0001$; ***$p < 0.001$; **$p < 0.01$; #$p < 0.0001$. Asterisks (*) refer to statistical comparisons between conditions and hashtags (#) to comparisons relative to basal values in each condition. Graphs represent mean ± SEM.

of CFA-triggered mechanical hypersensitivity but does not affect basal sensitivity.

Clinically, it is important to assess whether administration of PBN affects hypersensitivity after it has been fully established in inflammatory pain. We therefore delivered a single intrathecal dose of PBN two days after CFA injection (Fig. 6e) and observed that, within 6 h and lasting up to 24 h after PBN administration, mechanical hypersensitivity was significantly dampened (Fig. 6f, g).

Taken together, our findings reveal OAT1 to be a key mediator of persistent inflammatory hypersensitivity and a promising target for pain-relieving therapies.

## Discussion

In this study, we show that persistent nociceptive activity specifically drives HDAC4 out of the nucleus of neurons in the spinal cord dorsal horn and we causally link the subcellular localization of HDAC4 to the establishment of inflammatory hypersensitivity and to the regulated expression of pain-relevant genes. We identify the transporter OAT1 as an HDAC4-regulated gene and show it to be a functional target in spinal nociceptive plasticity. We thus describe a role for OAT1 in spinal neurons as a modulator of pain, and we link both the expression and activity of OAT1 expression to hypersensitivity.

The nucleo-cytoplasmic shuttling of class IIa HDACs members HDAC4, -5, -7, and -9 is an activity-dependent process in pyramidal neurons[18,19]. Surprisingly, in spinal cord neurons we found the activity-dependent export of class IIa HDACs to be specific for HDAC4. Mechanistically, a possible explanation for the distinct HDAC4 nuclear exclusion might be that HDAC4 is, so far, the only class IIa HDAC identified that bears a specific CaMKII-docking site and is phosphorylated by CDKL5[49]. Interestingly, both kinases have been associated with chronic pain states[50,51]. As such, they represent interesting putative signals that might drive specifically HDAC4 and no other class IIa HDACs out of the nucleus of dorsal horn neurons as observed in our study.

Accumulation of HDAC4 in the nucleus of neurons has been associated with several CNS disorders, including Alzheimer's disease, stroke, Parkinson's disease, retinal degeneration, and ataxia telangiectasia[52–55]. In contrast, our data show that under pathological pain conditions, HDAC4 in spinal cord neurons is preferentially confined to the cytosol. Therefore, what appears to be noxious in many neurodegenerative diseases, namely the nuclear localization of HDAC4, is seemingly beneficial in the context of chronic inflammatory pain for preventing the development of hypersensitivity. These observations can be reconciled if one considers that nuclear HDAC4 appears to oppose adaptations. Thus, in the case of physiological processes such as structural plasticity, survival, or memory formation—all of which require transcriptional flexibility—nuclear HDAC4 accumulation is harmful. By contrast, in the case of maladaptive phenomena such as addiction[56] or pathological pain, nuclear HDAC4 accumulation might instead be advantageous.

Reduced acetylation of histone 3 has been previously linked to the nuclear accumulation of HDAC4[54,57]. In agreement with this finding, we found that the activity-dependent nuclear export of HDAC4 in spinal cord neurons inversely correlates with elevated histone 3 acetylation levels, and that histone 3 acetylation could be prevented by exogenous expression of a nuclear localized dominant active HDAC4 mutant. An involvement of acetylation and HDACs in the signaling machinery underpinning pain has been hypothesized based on the fact that histone deacetylase inhibitors (HDACi) have shown analgesic properties in models of chronic pain[58–60]. Some studies, however, have come to contradictory conclusions based on results in which pain sensitivity was either enhanced following HDACs inhibition or ameliorated by the inhibition of histone acetyl transferases, enzymes that mediate histone acetylation[61–63]. Thus, it is not yet clear whether treatments aimed at reducing or increasing histone acetylation are expected to ameliorate or exacerbate chronic pain[23]. An additional complicating factor in the potential use of HDACi for pain treatment is the fact that the employed HDACi are often far from being specific for one particular HDAC[64]. Their use thus causes—at best—a broad and non-specific inhibition of all HDACs

belonging to one particular class or—most commonly—the inhibition of HDACs belonging to both classes I and II[58,64]. In such a scenario, it is virtually impossible to define the contribution of any single HDAC to pain chronicity and/or sensitivity. We took a different approach in this study. Driven by the input-dependent nature of some HDACs, we first characterized the impact of nociceptive activity on HDACs expression and subcellular localization, and identified HDAC4 as distinctively responsive in that it translocates from the nucleus to the cytoplasm in response to synaptic activity in primary spinal cord neurons or subsequent to the induction of peripheral inflammation in vivo. We then proceeded to specifically manipulate the relevant aspect—the subcellular localization—of this particular HDAC and thus verify its functional relevance for persistent nociceptive inflammatory hypersensitivity. Thus, although we cannot exclude the possibility that the activity of other HDACs might contribute to pathological signaling in chronic pain, we believe that nucleocytoplasmic translocation of HDAC4 plays a unique role in persistent inflammatory pain and that the specific modulation of its activity might be beneficial in pain therapy.

By preventing the nociceptive activity-dependent nuclear exclusion of HDAC4 specifically in spinal cord dorsal horn neurons, we were able to interfere with the development of the mechanical hypersensitivity that accompanies long-lasting inflammation. The development of thermal sensitivity was, however, not affected by this manipulation. Another study showed, by contrast, that the conditional knockout of HDAC4 in primary sensory neurons of mice affects peripheral sensitization and thermal hypersensitivity in the CFA model[20]. Therefore, a multi-faceted scenario involving HDAC4 in both the peripheral and central pain pathways is emerging. In peripheral sensory neurons, HDAC4 seems to promote the aspects of sensitization that are associated with thermal hypersensitivity, while centrally, in spinal cord neurons, its activity prevents development of mechanical hypersensitivity. The reasons for this dualistic behavior are most likely to be found in the genes controlled by HDAC4 that could differ between these compartments. Indeed, future comparative studies are needed to clarify this aspect of HDAC4's involvement in chronic pain. Notably, the differential effect of a given molecular signal such as HDAC4 on thermal or mechanical hypersensitivity is not an uncommon observation. Indeed, the literature offers plenty of examples showing a differential influence of specific signaling pathways on thermal and mechanical sensitivity in inflammatory pain[6,65–67], consistent with the idea that thermal and mechanical sensitization are modulated via the recruitment of distinct circuits or molecular signaling cascades.

In our study, we identified an HDAC4-regulated gene transcription program in inflammatory hypersensitivity that included several upregulated genes. This finding might seem surprising, as HDACs have traditionally been associated with the repression of transcription. HDAC4, however, has an extremely heterogeneous network of interacting proteins in both the cytosol and the nucleus, and appears to both induce and repress gene transcription[68]. Gene Ontology (GO) analysis of the differentially expressed genes (DEGs) after CFA delivery identified "glycine transport" in addition to the expected terms indicative of inflammatory responses. Surprisingly, the expression of Glyt1 (Slc6a9) or Glyt2 (Slc6a5), which are considered the main effectors in glycine transport, was left unchanged after CFA delivery. However, CFA significantly induced the expression of Slc6a20a, Slc36a2, and Slc38a5, which are listed under the GO term "glycine transport" but whose transport activity of glycine or other amino acids (i.e., proline, glutamine) is not fully characterized[22,23].

Among the identified genes regulated by HDAC4 we found OAT1 (Slc22a6). CFA triggered the upregulation of OAT1 in

dorsal horn neurons. The identity of the non-neuronal cells expressing OAT1 also at basal levels remains an open question. The mechanisms governing *Slc22a6* transcription are largely unknown. Of note, putative binding sites for myocyte enhancer factor-2 (MEF2) have been identified in the promoter region of human *SLC22A6*[69] and might explain the role of HDAC4 in the modulation of its transcription. MEF2 negatively regulates gene transcription by recruiting HDAC4 and associated co-repressor complexes while nuclear export of HDAC4 promotes MEF2 interaction with p300, stimulating transcription[70,71].

OAT1 is largely known for its classical role as a transporter of numerous molecules in renal tubular cells[24,72,73]. Although it has been detected in the brain, a clear role for OAT1 in physiological processes outside of the kidney has not yet been defined[44,74,75]. We describe that OAT1 expression in the spinal cord dorsal horn is induced in persistent inflammatory nociception and epigenetically regulated by HDAC4. Although HDAC4 modulates several genes, OAT1 downregulation in the CFA model appears to affect sensitization strongly. One likely explanation for the observed prominent effects of OAT1 manipulation is that, as OAT1 can transport multiple metabolites—and thereby affect many pathways—its altered expression might result in a cascade effect with consequences for many and varied signaling events. Nonetheless, although our results point to OAT1 as a prominent target of HDAC4, we cannot rule out the possibility that additional HDAC4 targets might play a role in sensitization.

OAT1 can transport prostaglandins and tryptophan metabolites[24,72,73], implicating this transporter within the CNS as a potential player in inflammatory and neuromodulatory processes. Our findings open up the exciting discussion as to which OAT1 substrates are mechanistically critical in spinal neurons for their role in sensitization. Although this remains to be systematically pursued in future studies, there are promising hints from the current literature. It has been suggested, for instance, that OAT1 mediates the transport of neuroactive metabolites away from the brain[44,76,77]. Several such substrates are linked to pain modulation, including prostaglandins (i.e., PGE2), but also tryptophan metabolites such as kynurenic acid (KYNA), and the serotonin metabolite 5-hydroxyindol acetate (5-HIAA)[78–82]. One substrate of OAT1 that has particular relevance for CNS function is KYNA, an endogenous antagonist of ionotropic glutamate receptors[83]. We hypothesize that OAT1 increases pain hypersensitivity via the modulation of inflammatory mediators like PGE2 and/or by promoting the clearance of KYNA from the spinal cord. In line with this thought, PBN, which inhibits the function of OAT1, amongst other targets, induces accumulation of KYNA within the CNS[84] and has analgesic properties in inflammatory pain models[85]. Our data, however, additionally indicate that the peripheral inflammation-triggered increase in OAT1 expression in the dorsal horn activates NF-kB, thus supporting the idea that OAT1 participates in the clearance of inflammatory signaling molecules. This possibility remains, however, an open question and a dynamic field of study to pursue in the future.

Taken together, we show that HDAC4 and OAT1 are key mediators of activity-dependent plasticity in the spinal cord and represent exciting targets for the treatment of inflammatory pain. Given that drugs specifically modulating HDAC4 are under development and especially in light of the fact that OAT1 can be pharmacologically modulated by drugs that are already clinically available, our findings open the way for innovative therapeutic interventions in chronic pain states. One possible delivery strategy for existing OAT1 modulators such as PBN could rely on an intrathecal pain pump to minimize doses and systemic side effects. On the other hand, the development of specific activators of HDAC4 that might prevent the establishment of epigenetically-regulated transcriptional changes in inflammatory pain states and, as such, interfere with the multiple diverse consequences of altered HDAC4 signaling, could possibly provide a more comprehensive and effective therapy.

## Methods

**Experimental model and subject details**. All animal procedures in this study were carried out in accordance with the ARRIVE guidelines and following approval by the local animal welfare committee (Regierungspräsidium, Karlsruhe, Germany; G3/19). Adult male or female C57BL/6N wild-type mice between 8 and 14 weeks of age (Charles River) were used throughout all in vivo experiments. For primary neuronal cultures, P0 C57BL/6N mice were used. Mice were housed in groups of maximally three animals in standard cages (15 cm × 21 cm × 13.5 cm) within approved animal facilities at Heidelberg University on a 12:12 h light:dark cycle and maintained at 45–65% humidity and 20–24 °C, and with ad libitum access to food and water. Housing conditions were constantly monitored.

**Recombinant adeno-associated virus production**. Recombinant adeno-associated viruses (rAAVs) serotype 1/2 were packaged and purified as described[10]. In brief, HEK293 cells (Stratagene, Cat# 240073; RRID:CVCL_6871) were grown in high-glucose-containing (4.5 g/liter) Dulbecco's Modified Eagle Medium (DMEM; Life Technologies) supplemented with 10% fetal bovine serum and antibiotics. Before transfection by standard calcium phosphate precipitation, medium was replaced with modified Dulbecco medium (IMDM; Life Technologies) containing 5% fetal bovine serum and no antibiotics. After transfection cells were returned to DMEM. Cells were collected at low speed centrifugation, resuspended in 150 mM NaCl-20 mM Tris-HCl and lysed by 0.5% sodium deoxycholate. rAAVs were purified on heparin affinity columns (HiTrap Heparin HP; GE Healthcare) and concentrated via Amicon Ultra-4 centrifugal filter devices (Millipore).

**Plasmids**. pAAV-LacZ, -HDAC4 wt, and -HDAC4 3SA plasmids and their respective rAAVs contain a Flag-tag at their C-terminal end and have been extensively characterized in vitro and in vivo for their specificity and efficacy[16,18]. Vectors used for shRNA-mediated knockdown contain a U6 promoter driving shRNA expression and an additional expression cassette encoding for GFP under the control of the CBA promoter (pAAV-shOat1-1, -shOat1-2, -shUNC). The following sequences, obtained from commercial siRNA targeting OAT1 (Qiagen) were used: CCCAGACAATTAAATAAATTT (shOat1-1), ATCCTAGT-GAATGGCATAATA (shOat1-2). As a control, a non-targeting shRNA sequence was used (shUNC)[10]. cDNA encoding for OAT1 was subcloned into a pAAV expression construct under the CMV/CBA promoter followed by an HA-tag at the C-terminal end to achieve overexpression of OAT1 (pAAV-OAT1). The genetically encoded calcium indicator jRGECO1a was subcloned in a pAAV expression construct under the CaMKIIα promoter and followed by a nuclear localization signal to generate pAAV-jRGECO1a-NLS.

**Small interfering RNAs**. A cocktail of four small interfering RNAs (siRNAs) targeting OAT1 (*Slc22a6*) or negative control siRNA (Qiagen) were intrathecally delivered using the in vivo-jetPEI® transfection reagent (Polyplus transfection) according to the manufacturer's instructions and established protocols[86]. siRNAs were intrathecally delivered once per day for three consecutive days. The following sequences were used: *Slc22a6*: AAGGAACTGACTCTAAACAAA (SI01420783), TCGGAAGGTGCTGATCTTGAA (SI01420790), ATCCTAGTGAATGGCA TAATA (SI01420797), CCCAGACAATTAAATAAATTT (SI01420804), and, as a negative control, AATTCTCCGAACGTGTCACGT (1022076).

**In vivo delivery of rAAVs**. Bilateral injection of rAAVs into the spinal parenchyma of adult mice allows for spatially- and temporally-restricted, stable gene transduction without causing persistent inflammation, tissue injury, or glial scar formation[6,10,87]. Mice were anesthetized with a mixture of fentanyl (50 µg/kg), midazolam (5 mg/kg), and medetomidine (0.5 mg/kg), and laminectomy was performed. Using a microprocessor-controlled minipump and a 35 gauge bevelled "NanoFil" needle (World Precision Instruments), 500 nl of a 2:1 mixture of rAAV stock with 20% mannitol were injected into the parenchyma of the spinal cord dorsal horn of the L3-L5 segments on each side (total of two injections per mouse) at a flow rate of 100 nl/min[6,10,87]. Anesthesia was reversed using a mixture of atipamezole (0.75 mg/kg), flumazenil (0.5 mg/kg), and naloxon (1.2 mg/kg), and the animals placed overnight on a heating plate set to 39 °C. Mice were allowed to recover for a minimum of 3 weeks after surgery before further analysis to allow for transgene expression. After experiments, mice were sacrificed, and correct delivery and expression or functionality of the injected rAAVs was assessed.

**Intrathecal injections**. Intrathecal injections were performed as previously described[86] and were used to deliver siRNAs (Qiagen) or probenecid (*p*-(di-*n*-propylsulfamyl) benzoic acid) (Sigma Aldrich). 50 mg probenecid were dissolved in a total volume of 5 ml solvent consisting of 3790 µl saline (0.9%), 850 µl Tris (1 M, pH 8.0), 150 µl NaOH (2 M), and 210 µl HCl (2 M) to yield a 10 µg/µl stock

solution. 16 µg of probenecid was delivered to each mouse intrathecally. The amount of probenecid injected—adjusted for the mouse cerebrospinal fluid volume—was determined based on the literature[88].

**Models of inflammatory pain**. For the induction of long-lasting inflammatory pain, 20 µl of Complete Freund's Adjuvant (CFA) (Sigma Aldrich) was sub-cutaneously injected into the plantar surface of one hindpaw[6,10]. Acute inflammatory pain was induced by injecting either 20 µl of 0.03% capsaicin (Tocris) or 1% formalin (Sigma Aldrich) solution in saline into the plantar surface of one hindpaw[6,10]. An equivalent volume of 0.9% saline was injected as control. All intraplantar injections were performed under inhalation anesthesia. Paw edema was verified via measurements of the paw width and thickness, obtained using a Micrometer Caliper (Fisher Scientific).

**Neuropathic pain model**. The spared nerve injury (SNI) model for neuropathic pain involving ligation of the tibial and common peroneal branches of the sciatic nerve has been described in detail previously[89]. Briefly, animals were anesthetized with a mixture of fentanyl (0.5 mg/kg), medetomidine (0.5 mg/kg), and midazolam (5.0 mg/kg). Once immobile, an ~1 cm incision was made in the skin overlying the biceps femoris muscle. The muscle was then bluntly dissected to reveal the sural, tibial, and common peroneal branches of the sciatic nerve. The tibial and common peroneal branches were ligated using 5/0 suture, and a 1–2 mm section of both nerves peripheral to the ligature removed. The wound was subsequently closed with sutures, and xylocain spray applied as a local anesthetic. Anesthesia was reversed using a mixture of atipamezole (0.75 mg/kg), flumazenil (0.5 mg/kg), and naloxon (1.2 mg/kg), and the animals placed overnight on a heating plate set to 39 °C. Animals were sacrificed 21–26 days following SNI surgery.

**Behavioral tests and relative analyses**. All behavioral tests were conducted by the same experimenter in awake and unrestrained mice during the light phase. Mice were acclimatized to the testing environment prior to behavioral measurements and on each day of testing. Baseline measures were obtained in separate sessions within one week prior to treatment. Mechanical sensitivity was assessed by measuring responses to paw pressure as previously described by applying a series of von Frey filaments (Ugo Basile, Aesthesio™) with ascending forces (0.008–2 g) to the plantar surface of the hindpaws. Each filament was applied five times to each paw in increasing order, starting with the filament producing the lowest force. To avoid repeated stimulus-induced sensitization, testing of both paws alternated between mice, and single paws were not re-stimulated before all other mice had been tested. Withdrawal frequency was calculated as a percentage of withdrawals out of the total number of applications for each von Frey filament. Sum of mechanical responses comprehensively represents data on response frequencies to graded 0.008–1.0 g von Frey hairs over the entire course of the behavioral experiment. Each filament was presented 5 times and in the case of 5 times withdrawal the percentage of responses to that filament would be 100%. The sum of percentage responses to 0.008–1.0 g von Frey forces was then calculated. The response frequency graphs are individual snapshots of sensitivity on the indicated day and depict the distinct percentage responses to the different filaments. Threshold was determined as the minimum von Frey force eliciting 40% of responses.

Thermal sensitivity was assessed according to the Hargreaves method using a plantar test analgesia meter (IITC Life Science)[6,10]. Following intraplantar capsaicin or formalin injections, mice were placed in an empty plexiglass chamber and the duration of all nocifensive behaviors, such as licking, shaking or flicking of the injected paw, were determined over a time period of 5 min after injection of capsaicin and over a time period of 60 min, divided into 5 min intervals, after formalin injection[6,10,37].

**Neuronal cultures and treatments**. Primary spinal cord and hippocampal cultures were plated and cultivated according to established protocols at 37 °C and 5% CO2[10,16,90]. Cultured neurons were transfected with 0.3 µg/ml plasmid encoding for pAAV-LacZ, -HDAC4 wt or HDAC4 3SA on day in vitro (DIV) 8 using Lipofectamine 2000 (Invitrogen) as described[16,90,91]. Cell death rates of cultured spinal cord neurons transfected LacZ, HDAC4 wt or HDAC4 3SA were determined by manually counting the number of collapsed or disintegrated nuclei of transfected neurons and dividing by the total number of transfected cells[91]. Primary cultured neurons were infected with rAAVs on DIV3 and maintained at 37 °C and 5% CO2 until analyses on DIV13/14. Cultured spinal cord neurons were treated with 50 µM bicuculline (Bic) to induce synaptic activity on DIV13/14 for 0.5 h, 2 h, or 8 h. Control cells were treated with vehicle only.

**Calcium imaging**. Nuclear calcium imaging was performed on DIV10 for primary spinal cord neurons infected with rAAV-jRGECO1a-NLS in a HEPES-buffered saline solution (HBS) containing, in mM: 140 NaCl, 2.5 KCl, 1.0 MgCl2, 2.0 CaCl2, 10 HEPES, 1.0 glycine, 35.6 D-glucose, and 0.5 Na-pyruvate. Fluorescence images were acquired from cells bathed in room-temperature HBS at 2.0 Hz with a cooled CCD camera (ImagEMX2, Hamamatsu) through a ×20 water-immersion objective (XLMPlanFluor, 0.95W, Olympus) on an upright microscope (BX51W1, Olympus). Fluorescence excitation (570 ± 10 nm; AHF Analysentechnik) was provided

by an LED light source (CoolLED pE-4000). jRGECO1a fluorescence was filtered using a 620 ± 30 nm emission filter (AHF Analysentechnik). Data were collected using proprietary software (VisiView, Visitron Systems), and analyzed using Fiji and IgorPro (WaveMetrics). Calcium signals in regions of interest drawn around individual nuclei were quantified as the baseline intensity-normalized change in fluorescence over time, $\Delta F/F = (F - F_{baseline})/F_{baseline}$. Responses to the GABAA receptor antagonist Bic (50 µM) were quantified in terms of the peak amplitude of the first transient and the mean inter-peak interval during the first five minutes following the start of Bic stimulation.

**QRT-PCR**. Total RNA was extracted from the dorsal part of the spinal cord or cultured neurons using an RNeasy Mini Kit (Qiagen), including optional DNase I treatment at room temperature for 15 min according to manufacturer's instructions[91,92]. Extracted RNA was reverse transcribed into first strand cDNA and quantitative reverse transcription PCR (QRT-PCR) was performed on a StepOne Plus real-time PCR system using TaqMan® gene expression assays (Applied Biosystems) for the following genes: *Gusb* (Mm00446953_m1), *Gapdh* (Mm99999915_g1), *Ptgs2* (Mm00478374_m1), *Slc22a6* (Mm00456258_m1), *Slc26a7* (Mm00524162_m1), *Prx* (Mm00479826_m1), *H19* (Mm01156721_g1), *Msln* (Mm00450770_m1), *Nov* (Mm00456855_m1) and *C1qc* (Mm00776126_m1). Expression of target genes was normalized against the expression of endogenous *Gusb* and *Gapdh* using StepOne software.

**RNA sequencing analysis**. Total RNA was isolated from the dorsal spinal cord tissue (L3–L5) of three individual mice per experimental condition. 0.3 µg of total RNA from each sample was used for RNAseq analysis. Integrity tests of RNA samples and differential gene expression analysis were performed by GATC Biotech (Inview Transcriptome Discover, GATC Biotech AG, Constance, Germany). Briefly, a strand-specific cDNA library was generated by purification and fragmentation of poly-A containing mRNA molecules, synthesis of strand-specific cDNA with random primers, adapter ligating on the cDNA and adapter-specific PCR amplification. Sequencing was performed with a Genome Sequencer Illumina HiSeq 4000 using 50 bp single end reads in FastQ format with at least 30 million reads (±3%). Gene expression was analyzed using the Bowtie, TopHat, Cufflinks, Cuffmerge, Cuffdiff software suite[93–95]. Mus musculus, mm10/GRCm38, Ensembl reference genome was used for transcript alignment. Genes were annotated using the v85 Ensembl database as reference. The cut-off for determining significantly differentially expressed genes (DEGs) was set to $p_{adjusted} < 0.05$ (false discovery rate adjusted p-value). Genes significantly regulated by CFA inflammation over the naïve state in saline injected mice, expressing either LacZ, HDAC4 wt or HDAC4 3SA, were considered upregulated with a log2 fold change > 0.5 and downregulated with a log2 fold change < −0.5. Gene ontology analysis of up- or downregulated DEGs in LacZ-expressing mice subjected to CFA or saline injection was performed using the PANTHER overrepresentation test (released 2020-04-07) and database (released 2020-02-21)[96]. Fisher's exact test with false discovery rate (FDR) correction was chosen and the background database was restricted to the pool of genes annotated in our RNA sequencing analysis. Data are available on GEO: GSE159895.

**Immunocytochemistry and quantification**. Cells were fixed for 20 min at room temperature (RT) with 4% paraformaldehyde and 4% sucrose in PBS (pH 7.4). Cells were washed three times with PBS and subsequently incubated for 3 h with primary antibodies (rabbit anti-AcH3 1:200 (Cell Signaling Cat# 9649), mouse anti-Flag 1:200 (Sigma Aldrich Cat# F3165 clone M2), rabbit anti-HA 1:200 (Santa Cruz Cat# sc-805), rabbit anti-HDAC1 1:200 (Thermo Scientific Cat# PA1-860), rabbit anti-HDAC3 1:200 (Cell Signaling Cat# 2632), rabbit anti-HDAC4 1:200 (Cell Signaling Cat# 7628), rabbit anti-HDAC5 1:200 (Cell Signaling Cat# 20458), rabbit anti-HDAC7 1:200 (Sigma Aldrich Cat# H2662), rabbit anti-HDAC9 1:200 (Abcam Cat# ab18970), rabbit anti-HDAC10 1:200 (Abcam Cat# ab53096), rabbit anti-HDAC11 1:200 (Abcam Cat# ab18973), mouse anti-NeuN 1:500 (Merck Cat# MAB377 clone 147714), rabbit anti-OAT1 1:200 (Abcam Cat# ab135924)) and for 1 h with appropriate secondary antibodies (1:400; goat-anti-rabbit IgG (H + L) Alexa Fluor 488 (Life Technologies Cat# A11008); goat-anti-mouse IgG (H + L) Alexa Fluor 594 (Life Technologies Cat# A11005)). All antibodies were diluted in GDB (0.1% gelatin, 0.3% Triton X-100, 15 mM Na2HPO4, 400 mM NaCl). Nuclei were visualized using Hoechst 33258 (2 µg/ml in 1x PBS). For analysis, up to eight random pictures from each condition were acquired using a Nikon Ni-E upright fluorescence microscope (NIS-Elements Software) or a Leica DM IRBE inverted fluorescence microscope with a ×40 objective. Intensity of nuclear fluorescence was measured in ImageJ using the Hoechst channel to define the nuclei as a region of interest in NeuN-positive cells. Measured intensity values were normalized to those obtained from vehicle-treated controls.

**Immunohistochemistry and quantification**. Mice were transcardially perfused with PBS followed by 10% formalin (Sigma Aldrich). Spinal cords and dorsal root ganglia (DRG) were isolated and post-fixed for 2 h in 10% formalin. Cryosections (20 µm) of the lumbar spinal cord segments L3–L5 and respective DRGs were mounted on Superfrost Plus Adhesion Microscope Slides™ (Thermo Scientific). Sections were blocked and permeabilized in blocking solution containing 10%

normal goat serum or normal donkey serum in 0.2% gelatin, 0.6% Triton X-100, 30 mM Na$_2$HPO$_4$, 800 mM NaCl and incubated with primary antibodies (rabbit anti-AcH3 1:1000, mouse anti-Flag 1:200, rabbit anti-HA 1:200, rabbit anti-HDAC1 1:200, rabbit anti-HDAC4 1:500, rabbit anti-HDAC7 1:200, rabbit anti-HDAC9 1:500, mouse anti-NeuN 1:1000, rabbit anti-OAT1 1:500, rabbit anti-cFos 1:1000 (Santa Cruz Cat# sc-52)) diluted in blocking solution overnight at 4 °C, and for 1.5 h at RT with secondary antibodies diluted 1:1000 in blocking solution. Slides were washed and incubated with 0.1% Sudan Black B (Sigma Aldrich) in 70% ethanol for 10 min at RT to reduce autofluorescence. For analysis, up to ten images of the dorsal horn area were acquired per condition using a Leica DM IRBE inverted fluorescence microscope with a ×20 objective. Intensity of nuclear fluorescence within neurons of the dorsal horn was measured in ImageJ using the Hoechst and NeuN channel to define the nuclei of neuronal cells as regions of interest. Intensity values were normalized to those from saline-injected controls. High-resolution images were acquired using a confocal laser microscope (Leica).

**Western blot analyses**. Spinal cords were quickly isolated and fresh tissue was rinsed in cold PBS before the dorsal part of lumbar spinal cord segments L3–5 was dissected and homogenized in RIPA buffer (150 mM NaCl, 1% Triton X-100, 0.57% sodium deoxycholate, 0.1% SDS, 50 mM Tris, pH 8, 1× Complete™ protease inhibitor cocktail (Roche)). Primary spinal cord cultures were lysed directly in sample buffer containing 30% glycerol, 4% SDS, 160 mM Tris-HCl, pH 6.8, and 0.02% bromophenol blue. For SDS-PAGE, 25 μg of spinal cord protein samples or 20 μl of primary culture lysates were used. Standard protocols for western blot were used. Relative protein content was normalized to loading controls. Images were generated using a Chemidoc Imaging system (Biorad). Antibodies used were: rabbit anti-AcH3 1:2000, rabbit anti-H3 1:4000 (Sigma Aldrich Cat# 06-755), mouse anti-Flag 1:500, rabbit anti-HA 1:1000, rabbit anti-HDAC1 1:2000, rabbit anti-HDAC4 1:6000, rabbit anti-HDAC6 1:1000 (Abcam Cat# ab1440), rabbit anti-HDAC7 1:2000, rabbit anti-HDAC9 1:4000, rabbit anti-OAT1 1:2000, rabbit anti-cFos 1:2000 (Santa Cruz Cat# sc-52), mouse anti-β-Actin 1:2000 (Santa Cruz Cat# sc-47778 clone C4), mouse anti-tubulin 1:400,000 (Merck Millipore Cat# T9026 clone DM1A), Goat anti-mouse IgG (H + L) Peroxidase 1:5000 (AffiniPure Jackson Immuno Research Cat# 115-035-003), Goat anti-rabbit IgG (H + L) 1:5000 Peroxidase (AffiniPure Jackson Immuno Research Cat# 111-035-144).

**Statistics and data analysis**. Data are presented as mean ± SEM and were analyzed using Microsoft Excel, Fiji, ImageJ. Statistical analyses were performed using GraphPad Prism 7 (GraphPad Software, Inc.). All data were tested for normality distribution. Details on the statistical tests used are indicated in the respective figure legends. All mice or dishes of cultured neurons were randomly assigned to the different experimental groups. All analyses were performed by a person blind to the condition used.

**Reporting summary**. Further information on research design is available in the Nature Research Reporting Summary linked to this article.

## Data availability

RNAseq data generated in this study have been deposited and are available on GEO: GSE159895. All other data presented in this study are provided in the Source Data file associated with this paper. Source data are provided with this paper.

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

## Acknowledgements

We are particularly grateful to Manuela Simonetti and Hilmar Bading for helpful discussions; Iris Bünzli-Ehret and Sara Lobato Moreno for their help with the preparation of neuronal cultures; Javier Sanchez Romero for general technical support; Daria Bunina and Deepti Mittal (project Z01 of CRC1158) for bioinformatic support; Anke Tappe-Theodor (project S01 of CRC1158) for her support with behavioral tests; and Prof. Michael Gekle for providing OAT1 cDNA. Some of the images were acquired at the Nikon Imaging Center at Heidelberg University. R. Kuner and D. Mauceri are members of the Heidelberg Pain Consortium (CRC1158) and HeiCINN at Heidelberg University, and R. Kuner is a member of the Molecular Medicine Partnership Unit with the European Molecular Biology Laboratory. This work was supported by the CRC1158 of the Deutsche Forschungsgemeinschaft (DFG; projects B01, B06, Z01 to R.K. and project A08 to D.M.).

## Author contributions

Investigation: C.L., E.P., J.L., A.M.H., A.K.K., and D.M.; Formal analysis: C.L. and D.M.; Visualization and writing: C.L. and D.M.; Resources, funding acquisition: R.K. and D.M.; Conceptual input: R.K. and D.M.; Conceptualization and design of the study: D.M.

## Funding

## Competing interests

The authors declare no potential conflicts of interest with respect to the research, authorship, and/or publication of this article. D. Mauceri is a founder and shareholder of FundaMental Pharma GmbH.
