## [Peer Review File · Nature Communications]

Reviewers' Comments:

Reviewer #1:

Remarks to the Author:

The authors investigated the role of the class IIa histone deacetylase 4 (HDAC4) and its potential target genes in complete Freund's adjuvant (CFA)-induced inflammatory nociception. Their results showed that total expression of different HDACs remained stable in primary neuronal cultures stimulated with the GABA antagonist bicuculline. However, nuclear HDAC4 levels decreased significantly associated with an increase in acetylated histone H3 indicating HDAC inactivation. A similar result was obtained in the spinal cord of mice which were injected with CFA into one hind paw indicating that long-lasting inflammatory nociceptive stimulation affects HDAC4 localization and activity in the spinal cord. Acute inflammatory nociception induced by capsaicin had no impact on HDACs. To further investigate the role of nuclear HDAC4 in nociception, the authors generated HDAC4 constructs with constitutive nuclear localization. After AAV-supported expression of mutant, wild type and control constructs in the dorsal horn of the spinal cord, it could be shown that CFA-induced AcH3 levels are significantly decreased in mice with the mutant HDAC4 construct. The behavioral analyses revealed a significant reduction of mechanical allodynia after CFA as well as inflammatory nociception in the second phase of the formalin test in mice treated with the mutated HDAC4 in comparison to the controls indicating that nuclear HDAC4 in dorsal horn neurons prevents central sensitization. Acute nociception and thermal hypersensitivity was not affected. With a transcriptome analysis, the authors identified OAT1 as potential target of HDAC4 and new player in the development of central sensitization. Pharmacological inhibition or RNA interference experiments targeting OAT1 indeed showed a reduction of mechanical allodynia after CFA while overexpression of OAT1 as proof of concept increased the nociceptive behavior. This is an interesting well-written manuscript providing data on a potential new pain-relevant protein which can be regulated by epigenetic mechanisms.

The study is timely and well-designed; however, there are a number of points that need to be overworked.

1. The authors showed shuttling of HDAC4 but not of other class II HDACs. This should be discussed in some more detail.
2. Page 6 line 127: The authors state that the HDAC4 regulation occurs specifically in neurons. The experiments shown have been performed in neuronal cultures, therefore this appears obvious. The sentence might be shifted to parts of the manuscript where HDAC4 expression in neurons has been shown in tissue samples.
3. Figure 1: c-fos expression is shown only as Western Blot and provided a double band after 2h CFA. Why? An immunohistochemical analysis would complete the experiment.
4. Figure 1e: The immunohistochemical staining for HDAC4 appears relatively weak, the signals in the CFA group appear rather unspecific. Background controls for the immunohistochemistry should be provided in the supplement.
5. Figure 1f: Please indicate which time points were integrated into the analysis.
6. In figure 2a, the vectors for the different HDAC constructs and the LacZ control are shown. They show a Flag Tag which is also described in the method section. However, suppl. Figure 3b indicates a GFP staining. Has a further vector been used for these analysis? Please clarify.
7. In figure 2b, it is relatively difficult to see differences in nuclear and cytosolic expression of HDAC4. A picture with a higher magnification might improve this.
8. Page 8, line 162: Heading is somewhat misleading since only one pain-relevant gene was investigated (Ptges).
9. In figure 3f, the authors show obviously strong regulations but do not indicate significance. Why? Although the p-values are provided in Suppl Table 1, it would be helpful in the figure as well.
10. The authors counted NeuN+ and NeuN- cells positive for OAT1 in figure 4 e, f, g. Then, in suppl fig.4, they showed that OAT1 is not expressed in astrocytes and glia cells. So, what kind of cells are the OAT1+NeuN- cells?
11. AcH3 and HDAC modification seems to occur on the ipsilateral side only. Is OAT1 expression altered ipsilaterally or also contralaterally? Why was the shRNA injected bilaterally?
12. Fig 5: 2 shRNAs have been used and it is indicated that they have significant impact on mechanical allodynia. From the statistical asterisks in figure 5g and h it is not clear if this applies for both shRNAs in each experiment or for only one. This should be presented in a clearer manner.

13. OAT1 expression after shRNA is only shown by PCR. An additional immunohistochemistry or Blot might be helpful. A control for overexpression of OAT1 is not provided.
14. Suppl Fig 5j is confusing. This part of the figure is not mentioned in the text, from color and legend it seems to belong to overexpression experiments but in the legend it is indicated as shRNA experiment. Please clarify.
15. Although the authors mentioned it in the discussion, it is not sufficiently explained why HDAC4 or OAT1 modulations do not affect thermal hyperalgesia.
16. The authors mention that nuclear expression of HDAC4 is dysregulated in neurodegenerative disease. From the context of the paper it would be interesting if there are also known dysregulations of OAT1 in these diseases.
17. Please provide the approval number for the animal experiments
18. It is indicated that formalin is injected into the plantar side of the paw. Is this right?

Reviewer #2:

Remarks to the Author:

The study by Litke and colleagues describes a role of HDAC4 and OAT1 in inflammation-induced pain. The authors show that HDAC4 is transported from nucleus to cytoplasm and using viral approach show that this process contributes to the development of mechanical hypersensitivity. They then focus on one of the HDAC4-regulated genes, OAT1, and demonstrate its role, using bidirectional approaches, in inflammatory pain. The study describes a novel gene expression mechanism in spinal cord neurons in inflammation. The experiments are well designed and presented. The paper is well written and is please to read.

In which cell types the nucleus levels of HDAC4 are reduced? Higher magnification images are needed to see the reduction in the nucleus and increase in cytoplasm.

Was overexpression of OAT1 confirmed at the protein level?

Given that the authors use SNI in this paper, it would be nice to see if HDAC4 localization is alerted in this model.

The behavioural effects of OAT1 manipulation (downregulation) in CFA seem to be similar to HDAC4 3SA, does it mean that it is the only or the main target? Maybe some discussion on this and other potential targets could be expanded.

Reviewer #3:

Remarks to the Author:

Review of the manuscript NCOMMS-21-13364

The authors report on the discovery of Oat1 as a mediator of persistent inflammatory pain. By applying plenty of techniques and mouse models the authors clearly show that HDAC4 and Oat1 are facilitating chronic pain. I want to congratulate the authors for this excellent piece of work especially for the discovery of a possible exciting new function of Oat1 in neurons. The study is well designed and executed, the methods are sound and a significant amount of evidence supports the conclusions. However, I have some comments the authors should consider:

1. One significant weak point of the study the authors did not mention nor discuss is the use of male mice for the study especially if the authors seriously considering this as a possible new therapy. It is known (Buist and Klassen, DMD, 2004) that Oat1 is expressed in a gender-dependent manner in rodents, especially the strain the authors have used. Oat1 is expressed 3-4-times higher in male mice. I am wondering if the authors would have been able to detect the connection between HDAC4 and Oat1 in female mice at all. I guess given the fact that the authors have already supplied a lot of evidence for the current story it would not be appropriate to ask for more studies in female mice. I would like to ask the authors to discuss this issue appropriately as a limiting factor of the study.
2. The second important point I want to make is that it looks odd that the authors show in figure 3b as a main functional change glycine transport but no change in glycine transporters or the glycine receptor shows up in the tables. This is especially interesting as glycine and the glycine receptor are important in the context of pain signaling (e.g. Hussein et al. Front. Mol. Neurosci., 2019, Arenas et al. J. Neuroinflammation, 2020). Of course, it could be post-transcriptional or

post-translational changes to glycine transporters or receptors facilitated by HDAC4 that would not show up in RNAseq analyses. Given the change in functional impact, the authors show in terms of glycine transport (10x higher than the organic anion transport) it would be nice to see if there is a change in expression of Glyt1 and Glyt2 which may further add to the inflammation but also pain. Glyt1 would be the main one for glial cells and Glyt2 the reuptake system in neurons.

3. Why is Ach3 different between figure 1f and 2d?

4. Does it not bother the authors that they have more than 50% genes upregulated in the LacZ control compared to HDAC4wt?

5. Fig. 3e+f: why do the bar graphs not show any error bars given 6 mice were analyzed? How many times was the qPCR done?

6. Fig. 4.a: The western blot for Oat1 needs a size note for the Oat1 band. In addition, the authors should show a western blot comparing neurons and kidneys to demonstrate that it is the full-length transporter they detect in neurons given the fact that splice forms for Oat1 have been identified.

7. Fig. 4b, d+h: error bars missing for saline or contralateral.

8. The size for all size bars of the pictures has to be included in the figure legend. And the F-values can be taken out of the figure legends as this is quite unusual.

9. In the discussion, the authors speculate that kynurenic acid (KYNA) would be cleared by Oat1 expressed in neurons. Well, that is very unlikely! KYNA is exclusively produced in astrocytes and works on e.g. NMDARs. So, do the authors think that any KYNA is produced in neurons or taken back up into neurons? Well, again most unlikely. More likely would be the scenario that neurons take up PGE2 from glial cells during or initiating further inflammation by further triggering the inflammasome. Would be interesting if the authors could show the activation of NFκB or NLRP3 in neurons to support their hypothesis of an inflammatory effect besides just the gene expression of inflammatory genes.

10. In a clinical setting if the authors could show this for patients, would the option then be to e.g. inject probenecid?

Dr. Andrew Bahn

The authors sincerely thank all reviewers for the highly constructive and helpful nature of the review and excellent suggestions. We have addressed all open questions and concerns and performed several new experiments, which have further validated and strengthened the main message of the manuscript. Changes made in the manuscript are marked in **bold text**.

Our detailed responses to the reviewers' comments and suggestions are as follows (the reviewers' original text is shown in italics):

Reviewer #1

***Comments:** The authors investigated the role of the class IIa histone deacetylase 4 (HDAC4) and its potential target genes in complete Freund's adjuvant (CFA)-induced inflammatory nociception. Their results showed that total expression of different HDACs remained stable in primary neuronal cultures stimulated with the GABA antagonist bicuculline. However, nuclear HDAC4 levels decreased significantly associated with an increase in acetylated histone H3 indicating HDAC inactivation. A similar result was obtained in the spinal cord of mice which were injected with CFA into one hind paw indicating that long-lasting inflammatory nociceptive stimulation affects HDAC4 localization and activity in the spinal cord. Acute inflammatory nociception induced by capsaicin had no impact on HDACs. To further investigate the role of nuclear HDAC4 in nociception, the authors generated HDAC4 constructs with constitutive nuclear localization. After AAV-supported expression of mutant, wild type and control constructs in the dorsal horn of the spinal cord, it could be shown that CFA-induced AcH3 levels are significantly decreased in mice with the mutant HDAC4 construct. The behavioral analyses revealed a significant reduction of mechanical allodynia after CFA as well as inflammatory nociception in the second phase of the formalin test in mice treated with the mutated HDAC4 in comparison to the controls indicating that nuclear HDAC4 in dorsal horn neurons prevents central sensitization. Acute nociception and thermal hypersensitivity was not affected. With a transcriptome analysis, the authors identified OAT1 as potential target of HDAC4 and new player in the development of central sensitization. Pharmacological inhibition or RNA interference experiments targeting OAT1 indeed showed a reduction of mechanical allodynia after CFA while overexpression of OAT1 as proof of concept increased the nociceptive behavior. This is an interesting well-written manuscript providing data on a potential new pain-relevant protein which can be regulated by epigenetic mechanisms. The study is timely and well-designed; however, there are a number of points that need to be overworked.*

Reply: Thank you for communicating your positive impression of the manuscript as being interesting, well-written, timely and well-designed and for raising important points, which we address below.

1. The authors showed shuttling of HDAC4 but not of other class II HDACs. This should be discussed in some more detail.

Reply: It is correct that HDAC4 was the only member of the diverse class IIa HDACs we analysed that displaying shuttling in our study, and we are happy to discuss this finding. Indeed, the observation that only HDAC4 displayed altered subcellular distribution as a result of inflammatory pain is a key aspect of our findings. Although it is true that HDAC4, -5, -7 and -9 can display similar

expression patterns and signal-induced subcellular distribution, it has been shown in multiple studies that they can have quite distinct characteristics.

The sequences of members of class IIa differ by 30-40% from each other, indicating potential functional differences. Moreover, a series of genetic studies determined that, although co-expressed, class IIa HDACs have widely distinct roles in different tissues. For instance, HDAC4 knockout mice are burdened by defects in osteogenesis, while mice lacking HDAC5 have exacerbated stress-induced cardiac hypertrophy and HDAC7 knockout mice fail to form tight junctions within the circulatory system¹. HDACs act at the top of genetic programs by influencing the genes expressed in a specific context; in spite of their homology, class IIa HDAC family members control largely non-overlapping transcriptional programs².

It therefore comes as no surprise that class IIa HDACs also exhibit different subcellular localization in identical physiological or pathological conditions. To name just a few examples, HDAC4 and HDAC5 have distinct nuclear localization patterns and functions in cocaine-related behaviours³; HDAC5 is almost entirely nuclear while HDAC4 is largely cytoplasmic in transformed fibroblasts; HDAC4 and HDAC5 have different cytosol/nuclear distribution in the hippocampus of patients affected by mesial temporal lobe epilepsy; HDAC5 and HDAC7 accumulate in the nucleus of retinal ganglion cells one day following an excitotoxic insult while HDAC4 does not⁴.

Mechanistically, answers as to why a particular class IIa HDAC might have a peculiar shuttling behaviour can be found by focusing on one of the key signals regulating nuclear import/export of these molecules, namely phosphorylation. Many studies have indeed shown that several kinases (i.e., CaMKII, PKD, LBK1, CDKL5) preferentially phosphorylate one HDAC and not its homologues⁵. In this regard, HDAC4 is, so far, the only class IIa HDAC that bears a specific CaMKII-docking site and is phosphorylated by CDKL5⁵. Interestingly, both kinases have been associated with chronic pain states^{6,7}, and therefore represent an interesting putative signal driving specifically HDAC4 out of the nucleus of dorsal horn neurons as observed in our study.

Following the reviewer's suggestions, a distilled version of these discussion points has now been included in the manuscript on page 17, lines 387-393.

2. Page 6 line 127: The authors state that the HDAC4 regulation occurs specifically in neurons. The experiments shown have been performed in neuronal cultures, therefore this appears obvious. The sentence might be shifted to parts of the manuscript where HDAC4 expression in neurons has been shown in tissue samples.

Reply: We understand the reviewer's suggestion of moving the statement on HDAC4 nuclear exclusion and increase in acetylation of histone 3 (ACh3) detected in cultured neurons to the part of our manuscript describing *in vivo* results. However, our primary cultures are not purely neuronal but also contain a small population of glial cells. Thus, the observations on HDACs subcellular localization and ACh3 level in neurons are not immediately obvious. We have amended the text to make this point explicit (page 6 lines 129-130).

3. *Figure 1: c-fos expression is shown only as Western Blot and provided a double band after 2h CFA. Why? An immunohistochemical analysis would complete the experiment.*

Reply: We appreciate the reviewer's points. Indeed, c-Fos is known to display multiple bands following stimuli as a result of different phosphorylation patterns ^{8, 9}. Moreover, different stimulation paradigms promote distinct patterns of the multiple bands. For instance, growth factor stimuli appear to result in more bands than depolarization ¹⁰.

We fully agree with the reviewer's suggestion of complementing our Western Blot data on c-Fos expression with an immunohistochemical analysis. We have now analysed c-Fos-positive cells in the dorsal horn of adult mice at different time points after CFA intraplantar injection. In agreement with previous literature ^{11, 12} and with our own Western Blot data (Suppl. Fig. 2l,m), we detected a significant increase in c-Fos-positive cells 2h after CFA injection. The new data are shown in Supplementary Figure 2j,k and mentioned on page 7 lines 157-159.

4. *Figure 1e: The immunohistochemical staining for HDAC4 appears relatively weak, the signals in the CFA group appear rather unspecific. Background controls for the immunohistochemistry should be provided in the supplement.*

Reply: We thank the reviewer for the suggestion about providing negative background controls for the immunohistochemistry experiments shown in figure 1e. Indeed, we routinely include negative controls without primary antibodies in all our histology analyses and have now included an example of the negative control performed in parallel to HDAC4 immunostaining in Supplementary Figure 2e.

5. *Figure 1f: Please indicate which time points were integrated into the analysis.*

Reply: Regarding the integrated analysis of the nuclear signal of HDAC4, HDAC7, HDAC9 and Ach3, the timepoints used were 0.5, 2, 6 and 24 h. We have included this information in the corresponding figure legend.

6. *In figure 2a, the vectors for the different HDAC constructs and the LacZ control are shown. They show a Flag Tag which is also described in the method section. However, suppl. Figure 3b indicates a GFP staining. Has a further vector been used for these analysis? Please clarify.*

Reply: We are happy to clarify about the nature of the construct used in supplementary figure 3b. As indicated in the corresponding figure legend, we injected the mice with rAAV-GFP. This was done to validate the correct localization, distribution and expression after intra-spinal delivery of rAAVs. This construct was chosen to further demonstrate the accuracy of our methodology to avoid additional potential artefacts resulting from antibody-based detection.

7. *In figure 2b, it is relatively difficult to see differences in nuclear and cytosolic expression of HDAC4. A picture with a higher magnification might improve this.*

Reply: We agree with the reviewer's view that including higher magnification images will help to better appreciate nuclear or cytosolic distribution of HDAC4 in Figure 2b. The requested images have now been added to Figure 2b.

8. Page 8, line 162: *Heading is somewhat misleading since only one pain-relevant gene was investigated (Ptges).*

Reply: In agreement with the reviewer's comment, we changed the heading to: **"Nuclear HDAC4 in spinal cord neurons alters the expression of the pain relevant gene Ptgs2 in vivo."**

9. *In figure 3f, the authors show obviously strong regulations but do not indicate significance. Why? Although the p-values are provided in Suppl Table 1, it would be helpful in the figure as well.*

Reply: Figure 3f was originally intended to compare magnitude and regulation of expression between RNAseq and qRT-PCR data. We now added to Figure 3e the significance of the RNAseq data, which are also shown in Table 1. To avoid redundancy, Figure 3f now only features qRT-PCR data and relative significance.

10. *The authors counted NeuN+ and NeuN- cells positive for OAT1 in figure 4 e, f, g. Then, in suppl fig.4, they showed that OAT1 is not expressed in astrocytes and glia cells. So, what kind of cells are the OAT1+NeuN- cells?*

Reply: We agree that it is important to clarify the cell type expressing OAT1. For quantification in Figure 4, we employed NeuN to mark nuclei of neurons, while the Hoechst dye labelled all nuclei making the identification and counting of single cells readily possible. There is about a 15% increase of OAT1+ cells over total cells (identified as Hoechst+) 24 h after CFA treatment (Fig. 4e) and a similar increase of OAT1+ neurons (identified as NeuN+, Fig. 4f). Therefore, we believe that CFA-mediated increase in OAT1 expression is happening in neurons, at least to the largest extent. This observation is further supported by the observation that we could fully prevent CFA-mediated induction of OAT1 using rAAV-HDAC4 3SA, which is only expressed in neurons due to tropism and the use of the neuron-specific *Synapsin* gene promoter (Figure 4i, m).

In order to uncover the identity of the non-neuronal cells expressing OAT1, we tried to perform a similar quantification for microglia cells, astrocytes and endothelial cells using anti-Iba1, -GFAP, -S100b or -CD-43 antibodies, respectively. We could not reach satisfactory conclusions due to technical limitations in spite of the numerous attempts. Thus, we cannot currently conclusively identify the non-neuronal cells expressing OAT1. We have acknowledged this limitation in the amended Discussion section on page 21, lines 475-477 and we amended and toned down the relative statement in the Results section page 13 lines 281-283.

11. *AcH3 and HDAC modification seems to occur on the ipsilateral side only. Is OAT1 expression altered ipsilaterally or also contralaterally? Why was the shRNA injected bilaterally?*

Reply: As the reviewer correctly noted, the induction of OAT1 expression takes place only ipsilaterally to the hindpaw injected with CFA, similar to our observations on induction of Ach3 and nuclear exclusion of HDAC4.

Nevertheless, we opted to perform bilateral injections of the shRNA-expressing virus in order to obtain data (histology, pain behaviour) on the possible effects of the injected constructs on the contralateral spinal cord and paw. This served as an internal control.

12. Fig 5: 2 shRNAs have been used and it is indicated that they have significant impact on mechanical allodynia. From the statistical asterisks in figure 5g and h it is not clear if this applies for both shRNAs in each experiment or for only one. This should be presented in a clearer manner.

Reply: As the reviewer correctly noted, the asterisks indicating statistically significant differences used in our original submission were not clearly and uniformly allocated to the relevant experimental conditions. We apologize for the confusion caused. To improve clarity, we have changed the way we indicated statistical significance not only in the figure mentioned by the reviewer (Fig. 5) but also in all other figures. In addition, the description of the symbols and the experimental conditions compared are included in the respective figure legends.

As for the specific question of the reviewer, those asterisks referred to the observation that both shRNA constructs gave statistically different changes as compared to the control construct in Fig. 5g, h.

13. *OAT1 expression after shRNA is only shown by PCR. An additional immunohistochemistry or Blot might be helpful. A control for overexpression of OAT1 is not provided.*

Reply: We agree with the reviewer's point that it would be helpful to substantiate the mRNA evidence for down-regulation of OAT1 (*Slc22a6*) by shRNA shown in our original submission with protein-related data. In agreement with the reviewer's request, Suppl. Fig. 5 now additionally shows that OAT1 protein level is reduced *in vitro* in cultured neurons infected with rAAV-shOat1-1 or with rAAV-shOat1-2 in comparison to rAAV-shUNC infected cultures (Suppl. Fig. 5f). Moreover, we also added data showing a reduction of OAT1 expression in the dorsal horn of mice spinally injected with rAAV-shOat1-1 or with rAAV-shOat1-2, in comparison to rAAV-shUNC-injected mice (Suppl. Fig 5k, l).

We are happy to provide the controls for overexpression of OAT1, as requested by the reviewer. We have now included representative immunocytochemistry images and Western blots from primary neuronal cultures infected with rAAV-LacZ or rAAV-OAT1 (Suppl. Fig. 6 a, b). rAAV-LacZ can be detected via its Flag tag while rAAV-OAT1 carries an HA tag. Moreover, we added representative images from immunohistological analysis of sections from the spinal cord dorsal horn of mice spinally injected with rAAV-LacZ or rAAV-OAT1 showing expression of the transgenes (Suppl. Fig. 6c). Finally, we provide Western Blot analysis from lysates of dorsal spinal cord sections L3-L5 of mice spinally injected with rAAV-LacZ or rAAV-OAT1 (Suppl. Fig. 6d).

14. *Suppl Fig 5j is confusing. This part of the figure is not mentioned in the text, from color and legend it seems to belong to overexpression experiments but in the legend it is indicated as shRNA experiment. Please clarify.*

Reply: We thank the reviewer for bringing this important error to our attention and we apologize for mistakes in description of the text as well as labelling of the figure. Indeed, the reviewer is correct, the data pertain to the overexpression of OAT1 but the legend was incorrectly referring to them as arising from an shRNA experiment. We have now corrected and simplified the figure. Three weeks after bilateral intraspinal delivery of either rAAV-LacZ or rAAV-OAT1, basal mechanical sensitivity (pre-inflammation) was significantly increased in OAT1-overexpressing mice compared to control mice expressing LacZ (Suppl. Fig. 6f).

15. *Although the authors mentioned it in the discussion, it is not sufficiently explained why HDAC4 or OAT1 modulations do not affect thermal hyperalgesia.*

Reply: The reviewer has raised a very good point. It has been suggested that mechanical and thermal hyperalgesia rely on the activation of different circuits and/or intracellular signaling pathways within the spinal cord. Indeed, there is sufficient evidence for differential mechanisms underlying the two types of sensitivity, and this differential influence is not uncommon^{12, 13, 14, 15}. Our data therefore place HDAC4 subcellular localization in dorsal horn neurons into the category of signaling events that mediate mechanical but not thermal sensitivity in inflammatory pain and add an important piece of information to the complex scenario of molecules and circuits determining hyperalgesia. We agree that it will be helpful to address this aspect in detail in future studies and have included it in the Discussion section (pages 20, lines 452-458).

16. *The authors mention that nuclear expression of HDAC4 is dysregulated in neurodegenerative disease. From the context of the paper it would be interesting if there are also known dysregulations of OAT1 in these diseases.*

Reply: We fully agree with the reviewer that this represents an interesting aspect. Although reports of OAT1 expression in the nervous system exist^{16, 17, 18}, they are few and mostly unrelated to neurodegenerative conditions. Up to now, only one study addressed the potential impact of OAT1 in neurodegeneration, specifically in Alzheimer's Disease (AD). The authors crossed OAT1 knockout mice with the AD mouse model tg2576 and found that deficits in OAT1 expression worsened the impairment in long-term-potential (LTP), learning and memory¹⁹. The role of OAT1 in the CNS and in neurodegenerative conditions therefore remains an unexplored line for future research, which we are currently actively pursuing.

17. *Please provide the approval number for the animal experiments*

Reply: In our original submission, the approval number was only included in the reporting summary. We thank the reviewer for bringing this point to our attention and we now additionally included the approval number in the method section.

18. It is indicated that formalin is injected into the plantar side of the paw. Is this right?

Reply: Formalin was subcutaneously injected, as indicated, into the plantar side of the hindpaw, as is standard for the formalin test^{20, 21}.

Reviewer #2

***Comments:** The study by Litke and colleagues describes a role of HDAC4 and OAT1 in inflammation-induced pain. The authors show that HDAC4 is transported from nucleus to cytoplasm and using viral approach show that this process contributes to the development of mechanical hypersensitivity. They then focus on one of the HDAC4-regulated genes, OAT1, and demonstrate its role, using bidirectional approaches, in inflammatory pain. The study describes a novel gene expression mechanism in spinal cord neurons in inflammation. The experiments are well designed and presented. The paper is well written and is please to read.*

Reply: We thank the reviewer for the positive comments and have addressed and clarified the open questions.

In which cell types the nucleus levels of HDAC4 are reduced? Higher magnification images are needed to see the reduction in the nucleus and increase in cytoplasm.

Reply: To better appreciate HDAC4 nuclear or cytosolic distribution, we were happy to follow the reviewer's suggestion and included the requested images in Suppl. Fig. 2f.

Was overexpression of OAT1 confirmed at the protein level?

Reply: Indeed, expression at protein level was tested in multiple ways. We routinely check all injected mice for correct transgene expression. We have now included representative immunocytochemistry images and Western blots from primary neuronal cultures infected with rAAV-LacZ or rAAV-OAT1 (Suppl. Fig. 6a,b). rAAV-LacZ can be detected via its Flag tag while rAAV-OAT1 carries an HA tag. Moreover, we added representative images from immunohistological analysis of sections from spinal cord dorsal horn of mice intraspinally injected with rAAV-LacZ or rAAV-OAT1 showing expression of the transgenes (Suppl. Fig. 6c). Finally, we provide Western Blot analysis from lysates of dorsal spinal cord sections L3-L5 of mice spinally injected with rAAV-LacZ or rAAV-OAT1, where we detected the expressed proteins at the expected molecular weight (Suppl. Fig. 6d).

Given that the authors use SNI in this paper, it would be nice to see if HDAC4 localization is alerted in this model.

Reply: We thank the reviewer for the constructive suggestion of investigating HDAC4 subcellular localization in the Spared Nerve Injury (SNI) model of neuropathic pain. In our original submission, we showed that OAT1 expression

in the dorsal horn was not affected by SNI (Fig. 4h). We have now analysed the nuclear localization of HDAC4 in dorsal horn neurons in the lumbar spinal cord segments L3-L5 after SNI (Suppl. Fig. 2i). We found no difference in HDAC4 nuclear content between sham and SNI conditions (Suppl. Fig. 2i). This result was further confirmed by differential analysis of the superficial (I-II) or deeper laminae (III-V) (Suppl. Fig. 2i). Taken together, these findings suggest that HDAC4-mediated regulation of OAT1 expression is unlikely to be involved in neuropathic pain subsequent to peripheral nerve injury.

The behavioural effects of OAT1 manipulation (downregulation) in CFA seem to be similar to HDAC4 3SA, does it mean that it is the only or the main target? Maybe some discussion on this and other potential targets could be expanded.

Reply: We agree with the reviewer that the phenotype obtained by OAT1 downregulation in the CFA model appears quite strong given the observation that several genes are affected by HDAC4 3SA in spinal neurons. As control mice injected with either control siRNA or rAAV-shUNC developed normal hypersensitivity, we believe that one can exclude any possible unspecific effects due to these procedures. One likely explanation for the observed prominent effects of OAT1 manipulation is that, as OAT1 can transport multiple metabolites and thereby affect many pathways, it might result in a cascade effect. Accordingly, interfering with OAT1 could potentially functionally impact a variety of different signaling events. A similar result was obtained in our past study¹², in which we manipulated either the entire nuclear calcium-dependent transcriptional program or single genes (i.e. Ptgs2, C1qc) and detected comparable impacts on spinal sensitization. Thus, although we believe that OAT1 is a prominent target of HDAC4, we cannot rule out the possibility that additional HDAC4 targets might have a functional impact on sensitization. The Discussion section has been expanded in relation to this aspect (pages 21-22, lines 488-496).

Reviewer #3

Comments: *The authors report on the discovery of Oat1 as a mediator of persistent inflammatory pain. By applying plenty of techniques and mouse models the authors clearly show that HDAC4 and Oat1 are facilitating chronic pain. I want to congratulate the authors for this excellent piece of work especially for the discovery of a possible exciting new function of Oat1 in neurons. The study is well designed and executed, the methods are sound and a significant amount of evidence supports the conclusions. However, I have some comments the authors should consider:*

Reply: We thank the reviewer for this positive assessment. The reviewer's comments are addressed in detail below.

1. One significant weak point of the study the authors did not mention nor discuss is the use of male mice for the study especially if the authors seriously considering this as a possible new therapy. It is known (Buist and Klassen, DMD, 2004) that Oat1 is expressed in a gender-dependent manner in rodents, especially the strain the authors have used. Oat1 is expressed 3-4-times higher in male mice. I am wondering if the authors would have been able to detect the connection between HDAC4 and Oat1 in female mice at all. I guess given the fact that the authors have already

supplied a lot of evidence for the current story it would not be appropriate to ask for more studies in female mice. I would like to ask the authors to discuss this issue appropriately as a limiting factor of the study.

Reply: The reviewer has raised the excellent point of potential differences in OAT1 expression between male and female rodents. We agree with this reviewer that this is an important issue and have now performed several experiments specifically addressing this aspect.

First, we have investigated whether intraplantar injection of CFA would affect HDAC4 subcellular localization in female mice. Indeed, we found that 24 h after CFA delivery, the nuclear levels of HDAC4 in dorsal horn neurons of adult female mice were reduced in a similar manner as what was previously detected for male mice (Fig. 4j). We next analysed OAT1 expression in the same mice. Similar to what was observed for male mice, following CFA injection, OAT1 expression was significantly induced in the dorsal horn of female mice in comparison to saline-injected mice (Fig. 4k). These results show that both HDAC4 and OAT1 behaved in a comparable manner in male and female mice in response to persistent inflammatory pain.

Finally, to verify the epigenetic control exerted by HDAC4 on OAT1 expression in female mice, we spinally injected rAAV-LacZ, -HDAC4 wild-type (wt) or -HDAC4 3SA in the dorsal L3-L5 segment of adult female mice. Three weeks after rAAV delivery, female mice were treated with CFA or saline and we assessed the development and course of mechanical hypersensitivity. Female mice spinally injected with either rAAV-HDAC4 3SA or control rAAVs showed similar basal sensitivity to mechanical stimuli applied to the hindpaws (Fig. 4l). At 24 h following CFA injection, all mice developed mechanical hypersensitivity (Fig. 4l). However, in comparison to control LacZ-expressing female mice, female mice expressing HDAC4 3SA in dorsal horn neurons showed significantly reduced mechanical hypersensitivity while those expressing HDAC4 wt did not differ from controls (Fig. 4l). After the behavioural analyses were completed, we sacrificed these female mice and analysed OAT1 expression. LacZ and HDAC4 wt-expressing female mice had higher levels of OAT1 resulting from CFA injection (Fig. 4m). In contrast, female mice expressing HDAC4 3SA showed no CFA-mediated induction of OAT1 (Fig. 4m). Taken together, our new results confirm the existence of a connection between persistent inflammatory pain, the subcellular localization of HDAC4 and the induction of OAT1 in the dorsal horn of mice of both sexes. Thus, although the literature indicates that OAT1 levels in the kidney at resting conditions are different between males and females, our data suggest that the inflammatory pain-triggered induction of OAT1 in the dorsal horn is not affected by sex differences.

These important new data are shown in Fig. 4j-m and described on page 13 lines 290- 301.

2. The second important point I want to make is that it looks odd that the authors show in figure 3b as a main functional change glycine transport but no change in glycine transporters or the glycine receptor shows up in the tables. This is especially interesting as glycine and the glycine receptor are important in the context of pain signaling (e.g. Hussein et al. Front. Mol. Neurosci., 2019, Arenas et al. J. Neuroinflammation, 2020). Of course, it could be post-transcriptional or post-translational changes to glycine transporters or receptors facilitated by HDAC4 that would not show up in RNAseq analyses. Given the change in functional impact, the

authors show in terms of glycine transport (10x higher than the organic anion transport) it would be nice to see if there is a change in expression of Glyt1 and Glyt2 which may further add to the inflammation but also pain. Glyt1 would be the main one for glial cells and Glyt2 the reuptake system in neurons.

Reply: The reviewer has correctly noticed that in our GO analysis the term “glycine transport” was mentioned but neither Glyt1 or Glyt2 were listed under Table 1 amongst the differentially expressed genes (DEGs). We are happy to address this apparent difference as described in detail below.

For our GO analysis, we specifically focused on the DEGs between LacZ saline and LacZ CFA conditions. Under the term “glycine transport” (<http://www.informatics.jax.org/go/term/GO:0015816>), 14 genes are listed (see table below). Of those, 3 genes (*Slc6a20a*, *Slc36a2*, *Slc38a5*) are differentially expressed between LacZ saline and LacZ CFA conditions and are present in Table 1. These 3 genes are still, however, not fully characterized in terms of their transport activity for glycine or other amino acids (i.e., proline, glutamine) ^{22, 23}

The reviewer is correct: Glyt1 (*Slc6a9*) and Glyt2 (*Slc6a5*) are not displayed in Table 1; neither shows a significant fold change following CFA-induced inflammation. In other words, the RNAseq analysis identified 3 out of 14 genes, which are associated with this GO term and that were at the same time present in our background gene list (LacZ Saline vs. LacZ CFA). However, Glyt1 and Glyt2 were not differentially expressed when comparing LacZ Saline vs CFA.

MGI Gene/Marker	Symbol	Name	Annotated Term	
MGI:1098271	Rgs2	regulator of G-protein signaling 2	negative regulation of glycine import across plasma membrane	
MGI:108409	Rgs4	regulator of G-protein signaling 4	negative regulation of glycine import across plasma membrane	
MGI:105090	Slc6a5	solute carrier family 6 (neurotransmitter transporter, glycine), member 5	glycine transport	
MGI:95760	Slc6a9	solute carrier family 6 (neurotransmitter transporter, glycine), member 9	glycine transport	
MGI:2442535	Slc6a17	solute carrier family 6 (neurotransmitter transporter), member 17	glycine transport	
MGI:2143217	Slc6a20a	solute carrier family 6 (neurotransmitter transporter), member 20A	glycine transport	identified in RNAseq data
MGI:1336891	Slc6a20b	solute carrier family 6 (neurotransmitter transporter), member 20B	glycine transport	
MGI:1355323	Slc7a8	solute carrier family 7 (cationic amino acid transporter, y+ system), member 8	glycine transport	
MGI:2384782	Slc25a38	solute carrier family 25, member 38	glycine import into mitochondrion	
MGI:1194488	Slc32a1	solute carrier family 32 (GABA vesicular transporter), member 1	glycine transport	
MGI:2445299	Slc36a1	solute carrier family 36 (proton/amino acid symporter), member 1	glycine transport	
MGI:1891430	Slc36a2	solute carrier family 36 (proton/amino acid symporter), member 2	glycine transport	identified in RNAseq data
MGI:2665001	Slc36a3	solute carrier family 36 (proton/amino acid symporter), member 3	glycine transport	
MGI:2148066	Slc38a5	solute carrier family 38, member 5	glycine transport	identified in RNAseq data

The following table summarizes the values and results of our RNA-seq analysis for Glyt1 (*Slc6a9*) and Glyt2 (*Slc6a5*) in all conditions. mRNA levels of Glyt1 (*Slc6a9*) did not significantly change in any of the analysed conditions. mRNA levels of Glyt2 (*Slc6a5*) appear significantly altered across some conditions but not when comparing LacZ Saline to LacZ CFA.

Condition	Gene	refseq_id	sample_1	sample_2	value_1	value_2	log2(fold_change)	foldchange	p_value	q_value	significant
1_lacZ_saline vs lacZ_CFA	Slc6a5 (Glyt2)	NM_00114601	LacZ_Saline	LacZ_CFA	95.1806	85.887	-0.148223	0.90236124	0.2367	0.9996	no
	Slc6a9 (Glyt1)	NM_008135	LacZ_Saline	LacZ_CFA	94.0985	95.147	0.0159932	1.01114731	0.924	0.9996	no
2_lacZ_saline vs 4wt_saline	Slc6a5 (Glyt2)	NM_00114601	LacZ_Saline	sample_4wt	97.089	58.514	-0.730534	0.6026808	5.00E-05	0.0009	yes
	Slc6a9 (Glyt1)	NM_008135	LacZ_Saline	sample_4wt	96.0028	71.812	-0.418848	0.74802169	0.0094	0.0859	no
3_lacZ_saline vs 3SA_saline	Slc6a5 (Glyt2)	NM_00114601	LacZ_Saline	sample_3SA	96.5766	66.842	-0.530925	0.69211084	5.00E-05	0.0016	yes
	Slc6a9 (Glyt1)	NM_008135	LacZ_Saline	sample_3SA	95.4948	73.534	-0.377008	0.7700329	0.024	0.2855	no
4_4wt_saline vs 3SA_saline	Slc6a5 (Glyt2)	NM_00114601	sample_4wt_ϕ	sample_3SA	59.246	68.394	0.207146	1.15440224	0.1113	0.9997	no
	Slc6a9 (Glyt1)	NM_008135	sample_4wt_ϕ	sample_3SA	72.7098	75.215	0.0488685	1.03445329	0.7896	0.9997	no
5_4wt_saline vs 4wt_CFA	Slc6a5 (Glyt2)	NM_00114601	sample_4wt_ϕ	sample_4wt	59.2246	59.981	0.0183072	1.01277044	0.8987	0.9983	no
	Slc6a9 (Glyt1)	NM_008135	sample_4wt_ϕ	sample_4wt	72.6862	82.29	0.179038	1.13212872	0.3456	0.9983	no
6_3SA_saline vs 3SA_CFA	Slc6a5 (Glyt2)	NM_00114601	sample_3SA_ϕ	sample_3SA	67.6832	52.747	-0.3597	0.77932662	0.0474	0.9996	no
	Slc6a9 (Glyt1)	NM_008135	sample_3SA_ϕ	sample_3SA	74.4551	72.801	-0.0324116	0.97778447	0.8988	0.9996	no
7_4wt_CFA vs 3SA_CFA	Slc6a5 (Glyt2)	NM_00114601	sample_4wt_ϕ	sample_3SA	59.5319	52.596	-0.178705	0.88349569	0.3343	0.9999	no
	Slc6a9 (Glyt1)	NM_008135	sample_4wt_ϕ	sample_3SA	81.6969	72.597	-0.170362	0.88861968	0.5033	0.9999	no
8_lacZ_saline vs. 4wt_CFA	Slc6a5 (Glyt2)	NM_00114601	LacZ_Saline	sample_4wt	2.89308	1.9384	-0.577759	0.67000371	0.655	0.9516	no
	Slc6a9 (Glyt1)	NM_008135	LacZ_Saline	sample_4wt	94.5917	81.218	-0.219912	0.85861781	0.1847	0.5251	no
9_lacZ_Saline vs. 3SA_CFA	Slc6a5 (Glyt2)	NM_00114601	LacZ_Saline	sample_3SA	96.1009	52.059	-0.884405	0.54171089	5.00E-05	0.0012	yes
	Slc6a9 (Glyt1)	NM_008135	LacZ_Saline	sample_3SA	95.0339	71.992	-0.400608	0.75753896	0.083	0.469	no
10_lacZ_CFA vs. 4wt_CFA	Slc6a5 (Glyt2)	NM_00114601	LacZ_CFA	sample_4wt	86.7867	58.821	-0.561149	0.67776216	5.00E-05	0.0011	yes
	Slc6a9 (Glyt1)	NM_008135	LacZ_CFA	sample_4wt	96.21	80.77	-0.252369	0.83951674	0.2045	0.6519	no
11_lacZ_CFA vs. 3SA_CFA	Slc6a5 (Glyt2)	NM_00114601	LacZ_CFA	sample_3SA	87.2267	51.736	-0.753596	0.59312332	5.00E-05	0.0015	yes
	Slc6a9 (Glyt1)	NM_008135	LacZ_CFA	sample_3SA	96.7274	71.52	-0.435572	0.73940054	0.0809	0.5301	no

The main focus of this study was the identification of those genes affected by nuclear HDAC4 under inflammatory conditions. Therefore, we first identified the DEGs between LacZ saline and LacZ CFA and then proceeded to investigate whether those genes were affected by HDAC4 wt and/or HDAC4 3SA expression.

Nevertheless, in agreement with the reviewer's suggestions, we further investigated expression of Glyt1 (*Slc6a9*) and Glyt2 (*Slc6a5*) both at the RNA and protein levels. First, we performed qRT-PCR analysis of dorsal spinal cords from mice spinally injected with rAAV-LacZ, -HDAC4 wt or -HDAC4 3SA 24h after either saline or CFA intraplantar injection. The results are shown in the figure below. We confirmed the RNA-seq data and found no difference in Glyt1 (*Slc6a9*) expression across any experimental conditions. Glyt2 (*Slc6a5*) mRNA levels were different between LacZ saline and HDAC4 wt saline and between LacZ saline and HDAC4 3SA saline conditions.

Further, we quantified the protein levels of Glyt1 and Glyt2 via immunohistological analysis of the dorsal horn of mice injected and treated as described above. The results are shown in the figure below. We found no difference for any condition for either Glyt1 or Glyt2.

(a,b) QRT-PCR analysis of *Slc6a5* or *Slc6a9* in lumbar L3-5 spinal cord segments 24 h after intraplantar injection of CFA or saline, in mice expressing LacZ, HDAC4 wt, or HDAC4 3SA (n=6 mice per condition). Expression values were normalized to those from rAAV-LacZ saline-injected mice and on *Gapdh* as housekeeping gene. (d,e) Quantification of relative fluorescence intensities of Glyt2 and Glyt1 the dorsal horn, following intraspinal rAAV delivery and intraplantar injections, as in (a,b) normalized to LacZ saline (n=3-4 mice per construct). Statistically significant differences were determined by one-way ANOVA followed by Tukey's test *, $p < 0.05$; In bar graphs, each point represents the mean value derived from one mouse. Graphs represent mean \pm SEM.

Taken together, our data demonstrate that Glyt1 and Glyt2 expression is not altered under inflammatory conditions or affected by nuclear HDAC4. Whether *Slc6a20a*, *Slc36a2*, *Slc38a5* affect preferentially glycine transport in the dorsal horn remains an open question which we believe falls beyond the scope of this study.

The Discussion section has been amended on pages 20-21 lines 465-473 to include an explanation on the point raised by the reviewer.

3. Why is ACh3 different between figure 1f and 2d?

Reply: Figure 1f represents an integration of ACh3 neuronal nuclear signal at different time points after CFA delivery (0.5, 2, 6 and 24 h). Figure 2d shows neuronal nuclear ACh3 signal only at the 24 h timepoint. As a result of the two different quantification and display methods, the data appear with different scales.

4. Does it not bother the authors that they have more than 50% genes upregulated in the LacZ control compared to HDAC4wt?

Reply: We thank the reviewer for their astute observation regarding the higher number of genes upregulated in HDAC4 wt-expressing mice compared to LacZ-expressing mice. HDAC4 wt retains biological activity and is capable of interacting with the numerous binding partners of endogenous HDAC4, both in the cytosol and in the nucleus of dorsal horn neurons. HDACs have been traditionally associated with repression of transcription via the recruitment of

repressors or by sequestering regulatory proteins in the cytosol. Class IIa HDACs, and HDAC4 in particular, however, have an extremely heterogeneous network of interacting proteins²⁴. There are several lines of evidence in the literature supporting the idea that HDAC4 activity can promote increases in gene expression modulated by particular factors such as FOXOs or STATs. A scenario in which HDAC4 is serving a multi-purpose function by both inducing and repressing gene transcription via its action on distinct pathways is thus emerging. It is likely that this accounts for the observed differences in gene regulation. We have now acknowledged this point in the discussion section (page 20, lines 460-465).

5. Fig. 3e+f: why do the bar graphs not show any error bars given 6 mice were analyzed? How many times was the qPCR done?

Reply: qRT-PCR was performed with 6 mice per condition (LacZ; HDAC4 3SA) or 5 mice per condition (HDAC4 wt, one mouse died before the 24h CFA timepoint). Figure 3f was originally intended to compare magnitude and regulation of expression between RNAseq and qRT-PCR data. We now added to panel 3e the significance of the RNAseq data, which are also shown in Table 1. To avoid redundancy, figure 3f now only features qRT-PCR data.

6. Fig. 4.a: The western blot for Oat1 needs a size note for the Oat1 band. In addition, the authors should show a western blot comparing neurons and kidneys to demonstrate that it is the full-length transporter they detect in neurons given the fact that splice forms for Oat1 have been identified.

Reply: We have added the size note in Fig. 4a for OAT1 as suggested by the reviewer. We have additionally performed Western blot analysis to compare lysates from dorsal spinal cord and kidney with two different anti-OAT1 antibodies (26574-1-AP from Proteintech and ab135924 from Abcam). As shown in the figure below, the detected band for the dorsal spinal cord, which is the one we analysed throughout our study, is similar in size to the one detected in kidney lysates.

Representative western blot analysis of lysates from the mouse kidney and the mouse dorsal spinal cord demonstrating that the size of the OAT1 band detected by two different antibodies has the same size in both tissue types.

7. Fig. 4b, d+h: error bars missing for saline or contralateral.

Reply: We thank the reviewer for noticing and we have added the error bars in revised Fig. 4.

8. The size for all size bars of the pictures has to be included in the figure legend. And the F-values can be taken out of the figure legends as this is quite unusual.

Reply: We agree and have included size bars for all images in all figures. F-values were included in figure legends as per journal instructions.

9. In the discussion, the authors speculate that kynurenic acid (KYNA) would be cleared by *Oat1* expressed in neurons. Well, that is very unlikely! KYNA is exclusively produced in astrocytes and works on e.g. NMDARs. So, do the authors think that any KYNA is produced in neurons or taken back up into neurons? Well, again most unlikely. More likely would be the scenario that neurons take up PGE2 from glial cells during or initiating further inflammation by further triggering the inflammasome. Would be interesting if the authors could show the activation of NFκB or NLRP3 in neurons to support their hypothesis of an inflammatory effect besides just the gene expression of inflammatory genes.

Reply: We understand the reviewer's doubts on the hypothesis that OAT1 might be involved in the clearance of kynurenic acid (KYNA). Among the numerous substrates of OAT1, KYNA has particular relevance for CNS function as it is an endogenous antagonist of glutamatergic ionotropic receptors with the highest affinity for the N-methyl-D-aspartate subtype of glutamate receptor (NMDAR)²⁵. NMDAR-mediated signaling is implicated in nearly all aspects of pain plasticity^{26, 27}. We therefore hypothesized that increased spinal cord levels of OAT1 in persistent inflammation may contribute to mechanical hypersensitivity by clearing spinal KYNA, thereby resulting in a disinhibition of NMDARs and an increase in excitatory synaptic transmission. To test this hypothesis, we first intrathecally administered either siOAT1 or siControl, followed by intraplantar CFA treatment in combination with intrathecal delivery of KYNA or vehicle (panel a, figure below). The quantity of kynurenic acid that was intrathecally delivered per mouse (5 μg) was selected based on existing literature on intrathecal delivery to mice that provided moderate analgesic effects in a model of chronic constriction injury²⁸. As high doses of KYNA might have full analgesic effects *per se*, we selected this lower dose to discern the potential effects of OAT1 expression. Consistent with the idea that OAT1 activity might limit KYNA availability in the spinal cord, KYNA treatment significantly reduced mechanical sensitivity in animals treated with siOat1, but not in animals that received siControl (panel b, figure below). Thus, mice with CFA-triggered increased levels of spinal OAT1 (siControl) displayed similar pain behavior independently of KYNA treatment while mice in which the induction of OAT1 was prevented by means of siOat1 were responsive to the analgesic effects of low dose KYNA. In other words, in control mice, the administered dose of KYNA might be too low to counteract CFA-induced mechanical hyperalgesia supported by the increased OAT1 levels which might promote KYNA clearance. In conditions when the CFA-mediated induction of OAT1 was prevented (siOat1), KYNA could display its mild analgesic effects possibly via slower clearance and/or prolonged availability. Our data therefore suggest that OAT1 in the dorsal spinal cord might modulate KYNA-mediated sensitization. We are aware, however, that these data are not fully exhaustive in explaining the regulation of OAT1-dependent KYNA fluxes and dynamics in the inflamed spinal cord and we therefore prefer to present them only in this letter and not in the manuscript.

(a) Timeframe of experiment and (b) quantification of mechanical sensitivity for mice intrathecally injected with the indicated siRNAs and KYNA or vehicle 6 h after intraplantar CFA injection, represented as 40% response threshold to mechanical force via von Frey filaments. Statistically significant differences were determined by two-tailed Student's *t*-test corrected by repeated measure with Holm-Sidak *, $p < 0.05$. Each point represents the value derived from one independent mouse ($n=6$ mice/condition). Graphs represent mean \pm SEM.

Since OAT1 might contribute to the clearance of inflammatory signaling molecules (i.e., Prostaglandin E₂), the reviewer further recommended looking at the activation of inflammation-related pathways such as the NF- κ B pathway. In agreement with the reviewer, we first quantified the levels of I κ B α in the dorsal spinal cord L3-L5 segment after intraplantar CFA injection. We found that I κ B α was significantly reduced 24h after CFA treatment (Suppl. Fig. 2I,m). I κ B α is a well-characterized inhibitory protein of NF- κ B that binds and traps NF- κ B in the cytoplasm. Upon stimulation, I κ B α gets degraded thus enabling NF- κ B to translocate to the nucleus and modulate transcriptional processes²⁹. CFA treatment therefore appears to promote the activation of NF- κ B via degradation of I κ B α . As shown in our manuscript, CFA triggers HDAC4 nuclear export and increases expression of OAT1 in the dorsal horn. To investigate whether increased OAT1 levels might activate the NF- κ B inflammatory pathway, we next quantified the levels of I κ B α in the dorsal spinal cord L3-L5 segments of mice spinally-injected with rAAV-LacZ or rAAV-OAT1 (Suppl. Fig. 6d,e). Over-expression of OAT1 in the dorsal spinal cord was sufficient to significantly decrease I κ B α levels (Suppl. Fig. 6d,e) similar to what was observed after CFA intraplantar injection (Suppl. Fig. 2I,m). Taken together, our results indicate that NF- κ B activation subsequent to an inflammatory stimulus may be supported by increased OAT1 expression in the dorsal horn.

It remains to be determined exactly which substrates of OAT1 would be affected and in which direction these might be transported. An *ad hoc* pharmacokinetic study aimed at the quantitative detection of OAT1 substrates in the spinal cord under different conditions and in the different compartments (i.e., CSF, parenchyma, blood)—and thus relying on technologically advanced and elaborate experimental settings—does not, in our view, belong to this study focused on the epigenetic regulation of persistent pain, but might represent material for follow-up investigations.

In addition to the data on I κ B α featured in the Results section (pages 7-8, lines 159-162; page 15, lines 335-338), we included in the Discussion a statement on the possible clearance of inflammatory molecules mediated by OAT1 in agreement with the reviewer's point of view and further supporting the concept of a still open scientific question (page 22-23 lines 514-517).

10. In a clinical setting if the authors could show this for patients, would the option then be to e.g. inject probenecid?

Reply: Regarding future potential therapeutic applications, probenecid-based therapies could indeed be a promising start. One of the key reasons we employed probenecid in this study is the fact that it is an FDA-approved drug already used in clinical practice. Accordingly, we specifically chose it to determine whether a drug that has already been deemed to be “safe” shows efficacy in our context. Indeed, analysis of the list of potential known OAT1 inhibitors shows that probenecid is, up to now, the only one additionally belonging to the category of already approved drugs potentially suitable for clinical applications (<http://transportal.compbio.ucsf.edu/transporters/SLC22A6/>) and, as such, corroborates the translational value of our findings.

In terms of administration, to minimize side effects, a viable idea could be the use of an intrathecal pump (“pain pump”), which features a reservoir to be implanted on the abdominal area between the muscle and skin of the patient and a catheter delivering medication to the spinal cord. The pump can last several years and can be programmed for delayed release of drugs over time. The drugs are directly delivered to the spinal cord and might be more efficient at lower doses than oral administration.

An interesting different approach for the future could be the modulation of HDAC4 activity that would prevent the establishment of epigenetically-regulated transcriptional changes in inflammatory pain, thus covering multiple aspects. At the moment, different HDAC4 inhibitors are under development, but an activator, and, moreover, an activator specific for HDAC4 is several steps away from being available.

A condensed version of these points can be found in the discussion section page 23 lines 525-532.

References cited in this document

1. Parra M. Class IIa HDACs - new insights into their functions in physiology and pathology. *Febs Journal* **282**, 1736-1744 (2015).
2. Sando R, 3rd, Gounko N, Pieraut S, Liao L, Yates J, 3rd, Maximov A. HDAC4 governs a transcriptional program essential for synaptic plasticity and memory. *Cell* **151**, 821-834 (2012).
3. Penrod RD, Carreira MB, Taniguchi M, Kumar J, Maddox SA, Cowan CW. Novel role and regulation of HDAC4 in cocaine-related behaviors. *Addict Biol* **23**, 653-664 (2018).
4. Schlüter A, Aksan B, Fioravanti R, Valente S, Mai A, Mauceri D. Histone Deacetylases Contribute to Excitotoxicity-Triggered Degeneration of Retinal Ganglion Cells In Vivo. *56*, 8018-8034 (2019).
5. Di Giorgio E, Brancolini C. Regulation of class IIa HDAC activities: it is not only matter of subcellular localization. *Epigenomics* **8**, 251-269 (2016).

6. Zhou YQ, *et al.* Cellular and Molecular Mechanisms of Calcium/Calmodulin-Dependent Protein Kinase II in Chronic Pain. *J Pharmacol Exp Ther* **363**, 176-183 (2017).
7. La Montanara P, *et al.* Cyclin-dependent-like kinase 5 is required for pain signaling in human sensory neurons and mouse models. *Sci Transl Med* **12**, (2020).
8. Chen RH, Abate C, Blenis J. Phosphorylation of the C-Fos Transrepression Domain by Mitogen-Activated Protein-Kinase and 90-Kda Ribosomal S6 Kinase. *P Natl Acad Sci USA* **90**, 10952-10956 (1993).
9. Murphy LO, Smith S, Chen RH, Fingar DC, Blenis J. Molecular interpretation of ERK signal duration by immediate early gene products. *Nature Cell Biology* **4**, 556-564 (2002).
10. Morgan JI, Curran T. Role of ion flux in the control of c-fos expression. *Nature* **322**, 552-555 (1986).
11. Hunt SP, Pini A, Evan G. Induction of c-fos-like protein in spinal cord neurons following sensory stimulation. *Nature* **328**, 632-634 (1987).
12. Simonetti M, *et al.* Nuclear calcium signaling in spinal neurons drives a genomic program required for persistent inflammatory pain. *Neuron* **77**, 43-57 (2013).
13. Mansikka H, Sheth RN, DeVries C, Lee H, Winchurch R, Raja SN. Nerve injury-induced mechanical but not thermal hyperalgesia is attenuated in neurokinin-1 receptor knockout mice. *Experimental Neurology* **162**, 343-349 (2000).
14. Peirs C, *et al.* Mechanical Allodynia Circuitry in the Dorsal Horn Is Defined by the Nature of the Injury. *Neuron* **109**, 73-90 e77 (2021).
15. Vardeh D, *et al.* COX2 in CNS neural cells mediates mechanical inflammatory pain hypersensitivity in mice. *Journal of Clinical Investigation* **119**, 287-294 (2009).
16. Alebouyeh M, *et al.* Expression of Human Organic Anion Transporters in the Choroid Plexus and Their Interactions With Neurotransmitter Metabolites. *Journal of pharmacological sciences* **93**, 430-436 (2003).
17. Cousins R, Wood CE. Expression of organic anion transporters 1 and 3 in the ovine fetal brain during the latter half of gestation. *Neurosci Lett* **484**, 22-25 (2010).
18. Bahn A, *et al.* Murine renal organic anion transporters mOAT1 and mOAT3 facilitate the transport of neuroactive tryptophan metabolites. *Am J Physiol Cell Physiol* **289**, C1075-1084 (2005).
19. Wu X, *et al.* Organic Anion Transporter 1 Deficiency Accelerates Learning and Memory Impairment in tg2576 Mice by Damaging Dendritic Spine Morphology and Activity. *J Mol Neurosci* **56**, 730-738 (2015).
20. Hunskaar S, Fasmer OB, Hole K. Formalin test in mice, a useful technique for evaluating mild analgesics. *Journal of neuroscience methods* **14**, 69-76 (1985).
21. Tjolsen A, Berge OG, Hunskaar S, Rosland JH, Hole K. The formalin test: an evaluation of the method. *Pain* **51**, 5-17 (1992).
22. Bhutia YD, Ganapathy V. Glutamine transporters in mammalian cells and their functions in physiology and cancer. *Bba-Mol Cell Res* **1863**, 2531-2539 (2016).

23. Kennedy DJ, Gatfield KM, Winpenny JP, Ganapathy V, Thwaites DT. Substrate specificity and functional characterisation of the H⁺/amino acid transporter rat PAT2 (Slc36a2). *Br J Pharmacol* **144**, 28-41 (2005).
24. Clocchiatti A, Di Giorgio E, Demarchi F, Brancolini C. Beside the MEF2 axis: unconventional functions of HDAC4. *Cell Signal* **25**, 269-276 (2013).
25. Stone TW. Neuropharmacology of quinolinic and kynurenic acids. *Pharmacol Rev* **45**, 309-379 (1993).
26. Woolf CJ, Salter MW. Neuroscience - Neuronal plasticity: Increasing the gain in pain. *Science* **288**, 1765-1768 (2000).
27. Luo C, Kuner T, Kuner R. Synaptic plasticity in pathological pain. *Trends Neurosci* **37**, 343-355 (2014).
28. Rojewska E, Ciapala K, Mika J. Kynurenic acid and zaprinast diminished CXCL17-evoked pain-related behaviour and enhanced morphine analgesia in a mouse neuropathic pain model. *Pharmacological reports : PR* **71**, 139-148 (2019).
29. Hatakeyama S, *et al.* Ubiquitin-dependent degradation of I kappa B alpha is mediated by a ubiquitin ligase Skp1/Cul 1/F-box protein FWD1. *P Natl Acad Sci USA* **96**, 3859-3863 (1999).

Reviewers' Comments:

Reviewer #1:

Remarks to the Author:

The authors have addressed all comments and suggestions from the first review round. I have no further recommendations.

Reviewer #2:

Remarks to the Author:

The authors have adequately addressed my concerns
Arkady Khoutorsky

Reviewer #3:

Remarks to the Author:

I would like to thank the authors for the way they have addressed my concerns. They have provided further exciting data and aspects beyond my request which have massively strengthened this manuscript. I have no further questions or concerns at this point and would like to congratulate the authors again for this excellent piece of work!

Manuscript Litke et al. (NCOMMS-21-13364A)

The authors sincerely thank all reviewers for the highly constructive and helpful nature of the review and excellent suggestions. All questions had been addressed in the first round of revisions. The current reviewer's comments (see below) require no further answers.

REVIEWERS' COMMENTS

Reviewer #1 (Remarks to the Author):

The authors have adressed all comments and suggestions from the first review round. I have no further recommendations.

Reviewer #2 (Remarks to the Author):

*The authors have adequately addressed my concerns
Arkady Khoutorsky*

Reviewer #3 (Remarks to the Author):

I would like to thank the authors for the way they have addressed my concerns. They have provided further exciting data and aspects beyond mt request which have massively strengthen this manuscript. I have no further questions or concerns at this point and would like to congratulate the authors again for this excellent piece of work!